# Biological learning in key-value memory networks

**Danil Tyulmankov**[*]
Columbia University
dt2586@columbia.edu

**Ching Fang**[*]
Columbia University
ching.fang@columbia.edu

**Annapurna Vadaparty**
Columbia University
Stanford University
apvadaparty@gmail.com

**Guangyu Robert Yang**
Columbia University
Massachusetts Institute of Technology
yanggr@mit.edu

## Abstract

In neuroscience, classical Hopfield networks are the standard biologically plausible model of long-term memory, relying on Hebbian plasticity for storage and attractor dynamics for recall. In contrast, memory-augmented neural networks in machine learning commonly use a key-value mechanism to store and read out memories in a single step. Such augmented networks achieve impressive feats of memory compared to traditional variants, yet their biological relevance is unclear. We propose an implementation of basic key-value memory that stores inputs using a combination of biologically plausible three-factor plasticity rules. The same rules are recovered when network parameters are meta-learned. Our network performs on par with classical Hopfield networks on autoassociative memory tasks and can be naturally extended to continual recall, heteroassociative memory, and sequence learning. Our results suggest a compelling alternative to the classical Hopfield network as a model of biological long-term memory.

## 1 Introduction

Long-term memory is an essential aspect of our everyday lives. It is the ability to rapidly memorize an experience or item, and to retain that memory in a retrievable form over a prolonged duration (days to years in humans). Neural networks capable of long-term memory have been studied in both neuroscience and machine learning, yet a wide gap remains between the mechanisms and interpretations of the two traditions.

In neuroscience, long-term associative memory is typically modeled by variants of Hopfield networks [Hopfield, 1982, Amit, 1992, Willshaw et al., 1969]. Rooted in statistical physics, they are one of the earliest and best known class of neural network models. A classical Hopfield network stores an activation pattern $\boldsymbol{\xi}$ by strengthening the recurrent connections $\boldsymbol{W}$ between co-active neurons using a biologically-plausible Hebbian plasticity rule,

$$\boldsymbol{W} \leftarrow \boldsymbol{W} + \boldsymbol{\xi}\boldsymbol{\xi}^{\mathsf{T}} \tag{1}$$

and allows retrieval of a memory from a corrupted version through recurrent attractor dynamics,

$$\boldsymbol{x}_{t+1} = \text{sign}(\boldsymbol{W}\boldsymbol{x}_t) \tag{2}$$

thereby providing a content-addressable and pattern-completing autoassociative memory.

In a more recent parallel thread in machine learning, various memory networks have been devised to augment traditional neural networks [Graves et al., 2014, Sukhbaatar et al., 2015, Munkhdalai et al.,

---

[*]Equal contribution

35th Conference on Neural Information Processing Systems (NeurIPS 2021).

2019, Le et al., 2019, Bartunov et al., 2019]. Memory-augmented neural networks utilize a more stable external memory system analogous to computer memory, in contrast to more volatile storage mechanisms such as recurrent neural networks [Rodriguez et al., 2019]. Many memory networks from this tradition can be viewed as consisting of memory slots where each slot can be addressed with a *key* and returns a memory *value*, although this storage scheme commonly lacks a mechanistic interpretation in terms of biological processes.

Key-value networks date back to at least the 1980s with Sparse Distributed Memory (SDM) as a model of human long-term memory [Kanerva, 1988, 1992]. Inspired by random-access memory in computers, it is at the core of many memory networks recently developed in machine learning [Graves et al., 2014, 2016, Sukhbaatar et al., 2015, Banino et al., 2020, Le et al., 2019]. A basic key-value network contains a key matrix $K$ and a value matrix $V$. Given a query vector $\widetilde{x}$, a memory read operation will retrieve an output $y$ as

$$\begin{aligned} h &= f(K\widetilde{x}) \\ y &= Vh \end{aligned} \quad (3)$$

where $f$ is an activation function that sparsifies the hidden response $h$.

Variations exist in the reading mechanisms of key-value memory networks. For example, $f$ may be the softmax function [Sukhbaatar et al., 2015], making memory retrieval equivalent to the "key-value attention" [Bahdanau et al., 2015] used in recent natural language processing models [Vaswani et al., 2017]. It has also been set as the step function [Kanerva, 1988] and hard-max function [Weston et al., 2015]. There is an even greater variation across writing mechanisms of memory networks. Some works rely on highly flexible mechanisms where an external controller learns which slots to write and overwrite [Graves et al., 2016, 2014], although appropriate memory write strategies can be difficult to learn. Other works have used simpler mechanisms where new memories can be appended sequentially to the existing set of memories [Sukhbaatar et al., 2015, Hung et al., 2019, Ramsauer et al., 2020] or written through gradient descent [Bartunov et al., 2019, Munkhdalai et al., 2019, Le et al., 2019, Krotov and Hopfield, 2016]. Kanerva [1992] updates the value matrix through Hebbian plasticity, but fixes the key matrix. Munkhdalai et al. [2019] turns memory write into a key-target value regression problem, and updates an arbitrary feedforward memory network using metalearned local regression targets.

A clear advantage of key-value memory over classical Hopfield networks is the decoupling of memory capacity from input dimension [Kanerva, 1992, Krotov and Hopfield, 2020]. In classical Hopfield networks, the input dimension determines the number of recurrent connections, and thus upper bounds the capacity. In key-value memory networks, by increasing the size of the hidden layer, the capacity can be much larger when measured against the input dimension (although the capacity per connection is similar).

There exists a gap between these two lines of research on neural networks for long-term memory – classical Hopfield networks in the tradition of computational neuroscience and key-value memory in machine learning. In our work, we study whether key-value memory networks used in machine learning can provide an alternative to classical Hopfield networks as biological models for long-term memory.

Modern Hopfield Networks (MHN) [Krotov and Hopfield, 2016, 2020, Ramsauer et al., 2020, Krotov, 2021] begin to address this issue, suggesting a neural network architecture for readout of key-value pairs from a synaptic weight matrix, but lacking a biological learning mechanism. MHNs implement an autoassociative memory (key is equal to the value, so $V = K^{\top}$) [Ramsauer et al., 2020, Krotov and Hopfield, 2020], but they can be used for heteroassociative recall by concatenating the key/value vectors and storing the concatenated versions instead. To query the network, keys can be clamped, and only the hidden and value neurons updated [Krotov and Hopfield, 2016]. With a particular choice of activation function and one-step dynamics, this architecture is mathematically equivalent to a fully connected feedforward network [Krotov and Hopfield, 2016], and thus analogous to the memory architecture proposed by Kanerva [1992].

It remains unclear whether a biological mechanism can implement the memory write. We will show that such biologically-plausible plasticity rules exist. We consider a special case of the MHN architecture and introduce a biologically plausible learning rule, using local three-factor synaptic plasticity [Gerstner et al., 2018], as well as respecting the topological constraints of biological neurons with spatially separated dendrites and axons. We first suggest a simplified version of the learning

rule which uses both Hebbian and non-Hebbian plasticity rules to store memories, and evaluate its performance in comparison to classical Hopfield networks. Adding increasing biological realism to our model, we find that meta-learning of the plasticity recovers rules similar to the simplified model. We finally show how the feedforward, slot-based structure of our network allows it be naturally applied to more biologically-motivated memory tasks. Thus, the addition of our learning rules makes key-value memory compatible with applications of the classical Hopfield network, particularly as a biologically plausible mechanistic model of long-term memory in neuroscience.

## 2 Simplified learning mechanism

We consider a neuronal implementation of slot-based key-value memory, and endow it with a biologically plausible plasticity rule for memorizing inputs. We first describe a simplified learning rule for storing key-value pairs, which sets an upper bound on the network memory capacity and serves as a benchmark against existing memory networks. In later sections, we modify this rule to further increase biological realism.

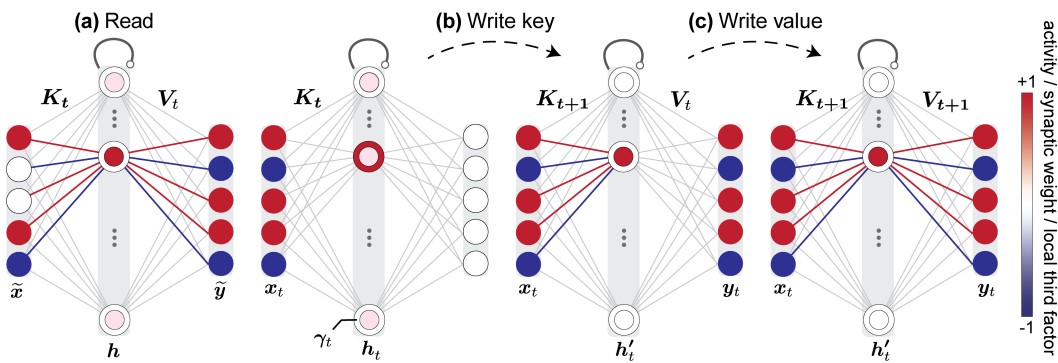

Figure 1: Network architecture and read/write mechanism. (a) Memory reading. The network is given a query $\widetilde{x}$ (input layer activation with a corrupted version of stored key $x$), selects the most similar stored key through approximately-one-hot hidden layer activity $h$, and returns the corresponding value $\widetilde{y} \approx y$. (b) Writing keys. The input $x_t$ is written into the $i^{\text{th}}$ slot of the input-to-hidden weight matrix $K_t$ by a "pre-only" plasticity rule, selecting the corresponding hidden neuron via local third factor $[\gamma_t]_i = 1$. (c) Writing values. The same $i^{\text{th}}$ hidden neuron is selected through an intermediate hidden unit activation $h'_t$ and the target $y_t$ is written to the hidden-to-output weight matrix $V_t$ by a Hebbian update.

The network operates sequentially and consists of three fully-connected layers of neurons (Figure 1): a $d$-dimensional input with activity at time $t$ given by the vector $x_t$, an $N$-dimensional hidden layer $h_t$, and an $m$-dimensional output layer $y_t$. The $i^{\text{th}}$ key memory slot corresponds to the synaptic weights from the input layer to a single hidden neuron ($i^{\text{th}}$ row of the key matrix $K_t$, storing key $x_i$). The corresponding value is stored in the weights from that hidden neuron to the output ($i^{\text{th}}$ column of the value matrix $V_t$, storing value $y_i$).

### 2.1 Reading

The network stores a set of key-value pairs $\{(x_i, y_i)\}$, such that if a query $\widetilde{x}$, e.g. a corrupted version of a stored key $x$, is presented to the network, it returns the corresponding value $y$. Given a query $\widetilde{x}$ (Figure 1a), the hidden layer computes its similarity $h_i$ to each stored key $x_i$ ($i^{\text{th}}$ row of $K_t$) as a normalized dot product:

$$h = \text{softmax}(K_t \widetilde{x}) \tag{4}$$

where the softmax function normalizes the hidden unit activations, and can be approximated biologically with inhibitory recurrent connections. The output layer then uses these similarity scores to compute the estimated value as the weighted sum of the stored values through a simple linear readout:

$$\widetilde{y} = V_t h = \sum_{i=1}^{N} h_i y_i \tag{5}$$

Assuming uncorrelated keys, the dot product of the query $\widetilde{x}$ will be maximal with its corresponding stored key $x$, and near-zero for all other stored keys, so $h$ will be approximately one-hot. Thus, Equation 5 reduces to $\widetilde{y} \approx y$ as desired. If target values are binary, $y_t \in \{+1, -1\}^m$ (neurons are either active or silent), we use $\text{sign}(\widetilde{y})$ when evaluating performance.

Mathematically, this architecture is equivalent to an MHN constrained to the heteroassociative setting with a fixed choice of activation functions [Krotov and Hopfield, 2020], and recurrent dynamics updated for exactly one step [Krotov and Hopfield, 2016]. Alternatively, it can be thought of as an differentiable version of an SDM [Kanerva, 1988] with a softmax rather than step function activation function in the hidden layer [Kanerva, 1992]. Unlike these networks, however, we introduce a novel biologically plausible writing mechanism to one-shot memorize key-value pairs by updating both the key and value matrices.

## 2.2 Writing keys

Key-value pairs are learned sequentially. Given an input $x_t$ at time $t$, we write it into slot $i$ of the key matrix using a non-Hebbian plasticity rule, where the presynaptic neuronal activity alone dictates the synaptic update, rather than pre- and postsynaptic co-activation as in a traditional Hebbian rule. A local third factor $[\gamma_t]_i \in \{0, 1\}$ (Figure 1b, red circle) gates the plasticity of all input connections to hidden unit $i$, enabling selection of a single neuron for writing. Biologically, this may correspond to a dentritic spike [Stuart and Spruston, 2015, Gambino et al., 2014], which occur largely independently of the somatic activity $h_t$ performing feedforward computation[2]. This plasticity rule resembles behavioral time scale plasticity (BTSP) [Bittner et al., 2017], recently discovered in hippocampus, a brain structure critical for formation of long-term memory.

In this simplified version we approximate local third factors as occurring in the least-recently-used neuron by cycling through the hidden units sequentially:

$$[\gamma_t]_i = \begin{cases} 1 \text{ if } t = i \bmod N \\ 0 \text{ otherwise} \end{cases} \tag{6}$$

This can be biologically justified in several ways. Each neuron may have an internal timing mechanism that deploys a local third factor every $N$ timesteps; the neurons may be wired such that a dendritic spike in neuron $i$ primes neuron $i + 1$ for a dendritic spike at the next time step; or the local third factors may be controlled by an external circuit that coordinates their firing. Alternatively, we consider a simpler mechanism, not requiring any coordination among the hidden layer neurons: each neuron independently has some probability $p$ of generating a dendritic spike:

$$[\gamma_t]_i \sim \text{Bernoulli}(p) \tag{7}$$

To gate whether a stimulus should be stored at all, we include a scalar global third factor $q_t \in \{0, 1\}$. Biologically, this may correspond to a neuromodulator such as acetylcholine [Rasmusson, 2000] that affects the entire population of neurons, controlled by novelty, attention, or other global signals. Although it can also be computed by an external circuit, in our experiments this value is provided as part of the input. Thus, the learning rate of the synapse between input unit $j$ and hidden unit $i$ is the product of the local and global third factors:

$$[\eta_t^{\text{K}}]_{ij} = q_t [\gamma_t]_i \tag{8}$$

Finally, to allow reuse of memory slots, we introduce a forgetting mechanism, corresponding to a rapid synaptic decay mediated by the local third factor. Whenever a synapse gets updated we have $[\eta_t^{\text{K}}]_{ij} = 1$, so we can zero out its value by multiplying it by $1 - [\eta_t^{\text{K}}]_{ij}$. If there is no update (either third factor is zero), the weight is not affected. The synaptic update is therefore:

$$K_{t+1} = (1 - \eta_t^{\text{K}}) \odot K_t + \eta_t^{\text{K}} \odot [\mathbf{1} x_t^T] \tag{9}$$

where $\odot$ indicates the Hadamard (element-wise) product, and $\mathbf{1} \equiv (1, 1, .., 1)$.

---

[2]The same synaptic update could be accomplished without third factors, using a Hebbian rule, if the hidden layer were one-hot. However, since the hidden layer activity is determined by its weight matrix and the input, there is no simple neuronal mechanism to independently select a hidden neuron. More complicated versions may involve an external circuit which controls the activity of the hidden layer during writing, or strong feedforward weights which do not undergo plasticity [Tyulmankov et al., 2021].

## 2.3 Writing values

Having stored the key $\boldsymbol{x}_t$, the hidden layer activity at this intermediate stage is given by:

$$\boldsymbol{h}'_t = \mathrm{softmax}(\boldsymbol{K}_{t+1}\boldsymbol{x}_t) \tag{10}$$

Note we are using the *updated* key matrix with $\boldsymbol{x}_t$ stored in the $i^{\text{th}}$ slot. In the idealized case of random uncorrelated binary random memories $\boldsymbol{x}_t \in \{+1, -1\}^d$, the $i^{\text{th}}$ entry of $\boldsymbol{K}_{t+1}\boldsymbol{x}_t$ will be equal to $\boldsymbol{x}_t \cdot \boldsymbol{x}_t = d$. All other entries will be near 0, since they are uncorrelated with $\boldsymbol{x}_t$. Thus, after normalization via the softmax, $\boldsymbol{h}'_t$ will be approximately one-hot, with $[\boldsymbol{h}'_t]_i \approx 1$.

To store the value, the output layer activity is clamped to the target $\boldsymbol{y}_t$ (Figure 1c). Biologically, this can be achieved either by strong one-to-one residual connections from the input to the output layer, or by having a common circuit which drives activity in both the input and output layers. Since the only hidden unit active is the one indexing the $i$th column of the value matrix, we can update the synapse between hidden unit $i$ and output unit $k$ by a Hebbian rule with learning rate $[\boldsymbol{\eta}^{\text{v}}_t]_{ki} = q_t[\boldsymbol{\gamma}_t]_i$ and rapid decay rate analogous to the key matrix:[3]

$$\boldsymbol{V}_{t+1} = (1 - \boldsymbol{\eta}^{\text{v}}_t) \odot \boldsymbol{V}_t + \boldsymbol{\eta}^{\text{v}}_t \odot \boldsymbol{y}_t(\boldsymbol{h}'_t)^T \tag{11}$$

# 3 Results

## 3.1 Benchmark: autoassociative recall

As a simple evaluation of our algorithm's performance and comparison to other memory networks, we consider the classical autoassociative memory recall task where the key is equal to the value, $(d = m,$ Figure 1). The network stores a set of $T$ stimuli $\{\boldsymbol{x}_t\}$ and the query $\widetilde{\boldsymbol{x}}$ is a corrupted version of a stored key (60% of the entries are randomly set to zero). The network returns the corresponding uncorrupted version (Appendix A, Figure A1).

We compute the accuracy as a function of the number of stored stimuli (Figure 2a). By design, the plasticity rule with sequential local third factors performs identically to a simple non-biological key-value memory (TVT, [Hung et al., 2019], Appendix B). It has perfect accuracy for $T \leq N$ since every pattern is stored in a unique slot. For $T > N$, previously stored patterns are overwritten one-by-one and accuracy smoothly degrades. With random local third factors, accuracy degrades sooner due to random overwriting. Importantly, this holds for arbitrary network sizes, unlike the classical Hopfield network, which experiences a "blackout catastrophe" where all stored patterns are erased if the network size is large and the number of inputs exceeds its capacity [McCloskey and Cohen, 1989, Robins and McCallum, 1998] (we do not see this here due to a relatively small network size).

Next, by measuring the number of patterns that the network can store before its recall accuracy drops below a threshold ($\theta = 0.98$), we can estimate this network's memory capacity $C$. This is an imperfect measure because some networks may have accuracy that stays above threshold longer but drops sharply after that. Nevertheless, this metric allows us to investigate empirically how the network's performance scales with its size (Figure 2b). The empirical scaling of the classical Hopfield network is $C \sim 0.14N$, consistent with theoretical calculations [Amit et al., 1985], and the sequential algorithm's capacity scales approximately as $C \sim 1.0N$, as expected analytically. Importantly, the random algorithm's capacity also scales linearly, $C \sim 0.16N$, with a slope similar to the Hopfield network. Note, however, that with the same number of hidden neurons our network has twice as many connections as the Hopfield network. We also note that although the theoretical capacity of the Hopfield network can scale as $C \sim 2N$ [Cover, 1965, Gardner, 1988], to our knowledge there is not a learning algorithm that achieves this bound. Other work on autoassociative memory shows exponential scaling, but lacks a biologically plausible readout [Demircigil et al., 2017].

## 3.2 Meta-learning of plasticity rules

We now introduce parameters that can be meta-learned to optimize performance on a particular dataset without sacrificing biological plausibility. First, we enable varying the scale of the synaptic

---

[3] If local third factors are dendritic spikes in the *dendrites* of hidden layer neurons, it may not be realistic for the *axons* of the same neurons to be affected by these dendritic spikes. We consider this case for simplicity, as an upper-bound on performance. We will later lift this assumption without significantly hurting performance.

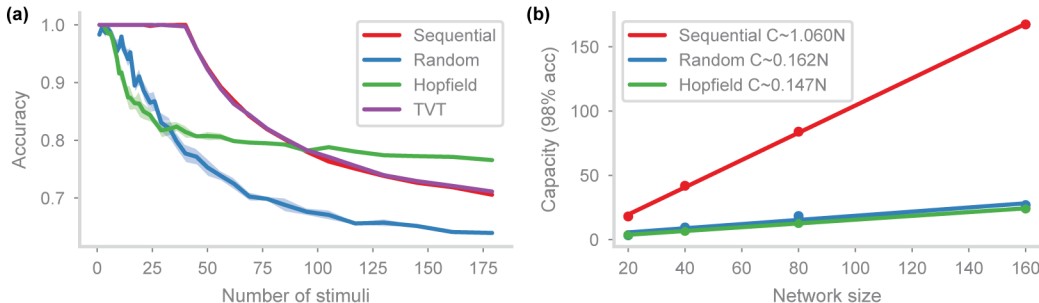

Figure 2: Our network ($d = N = 40$) with sequential and random ($p = 0.1$) local third factors, Hopfield network, and TVT [Hung et al., 2019] performance on the autoassociative memory task. (a) Accuracy as a function of stored stimuli. (b) Capacity (maximum number of stored stimuli at $\geq 98\%$ accuracy) as a function of network size.

plasticity rates – each one is multiplied by a learnable parameter $\widetilde{\boldsymbol{\eta}}^{\text{K}}, \widetilde{\boldsymbol{\eta}}^{\text{V}}$:

$$[\boldsymbol{\eta}_t^{\text{K}}]_{ij} = q_t[\boldsymbol{\gamma}_t]_i[\widetilde{\boldsymbol{\eta}}^{\text{K}}]_{ij} \text{ and } [\boldsymbol{\eta}_t^{\text{V}}]_{ki} = q_t[\widetilde{\boldsymbol{\eta}}^{\text{V}}]_{ki} \tag{12}$$

Next, as a further improvement on biological plausibility, we remove the assumption that the hidden-to-output synapses (hidden layer axons) have access to the local third factors (hidden layer dendritic spikes). Note that $\boldsymbol{\eta}_t^{\text{V}}$ above no longer contains a $\boldsymbol{\gamma}_t$ term. Instead, as a forgetting mechanism, the hidden-to-output synapses decay by a fixed factor $\widetilde{\boldsymbol{\lambda}}$ each time a stimulus is stored:

$$\boldsymbol{\lambda}_t = (1 - q_t) + q_t\widetilde{\boldsymbol{\lambda}} \tag{13}$$

where $\boldsymbol{\lambda}_t$ is the decay at time $t$, and $q_t \in \{0, 1\}$. Although in general $\widetilde{\boldsymbol{\eta}}^{\text{K}}, \widetilde{\boldsymbol{\eta}}^{\text{V}}$, and $\widetilde{\boldsymbol{\lambda}}$ can each be a matrix which sets a learning/decay rate for each synapse or neuron, we consider the simpler case where each is a scalar, shared across all synapses.

Most importantly, we parameterize the update rule itself. Each of the pre- and postsynaptic firing rates is linearly transformed before the synaptic weight is updated according to the prototypical "pre-times-post" rule. The $j$th input neuron's transformed firing rate is given by

$$[f^{\text{K}}(\boldsymbol{x})]_j = \widetilde{a}^{f^{\text{K}}}x_j + \widetilde{b}^{f^{\text{K}}} \tag{14}$$

and others $f^{\text{V}}, g^{\text{K}}, g^{\text{V}}$ are analogous. Although these functions can take arbitrary forms in general, this simple parameterization enables interpolating between traditional Hebbian, anti-Hebbian, and non-Hebbian (pre- or post-only) rules. The update rules can be summarized as follows:

$$\boldsymbol{K}_{t+1} = (1 - \boldsymbol{\eta}_t^{\text{K}}) \odot \boldsymbol{K}_t + \boldsymbol{\eta}_t^{\text{K}} \odot [g^{\text{K}}(\boldsymbol{h}_t)f^{\text{K}}(\boldsymbol{x}_t)^T] \tag{15}$$

$$\boldsymbol{V}_{t+1} = \boldsymbol{\lambda}_t \odot \boldsymbol{V}_t + q_t\widetilde{\boldsymbol{\eta}}^{\text{V}} \odot [g^{\text{V}}(\boldsymbol{y}_t)f^{\text{V}}(\boldsymbol{h}_t')^T] \tag{16}$$

We begin by training this network on the benchmark task from subsection 3.1, optimizing these parameters using stochastic gradient descent (Adam, [Kingma and Ba, 2014]). Figure 3a shows the performance of the meta-learned algorithms in comparison to the corresponding simplified versions (section 2) for either sequential and random local third factors. Importantly, removing the unrealistic local-third-factor-mediated rapid synaptic decay in the hidden-to-output synapses and replacing it with a passive decay does not significantly impact performance, as long as the decay rate is appropriately set. Indeed, this even slightly improves performance for longer sequences.

In the case of a random local third factor, we also compare values of $p$ (not trained). Empirically, we find that $p \approx 4/N$ produces desirable performance: it ensures that the probability of no local third factor occurring (and therefore no storage) is small ($< 2\%$) while minimizing overwriting. Computing the capacity (Figure 3b), we see that there is an advantage for the probability to scale with the network size ($p = 4/N$) rather being a fixed value ($p = 0.1$).

We next investigate the training trajectories. Figure 3c shows the loss and accuracy curves over the course of training for networks with sequential and random local third factors. Training data

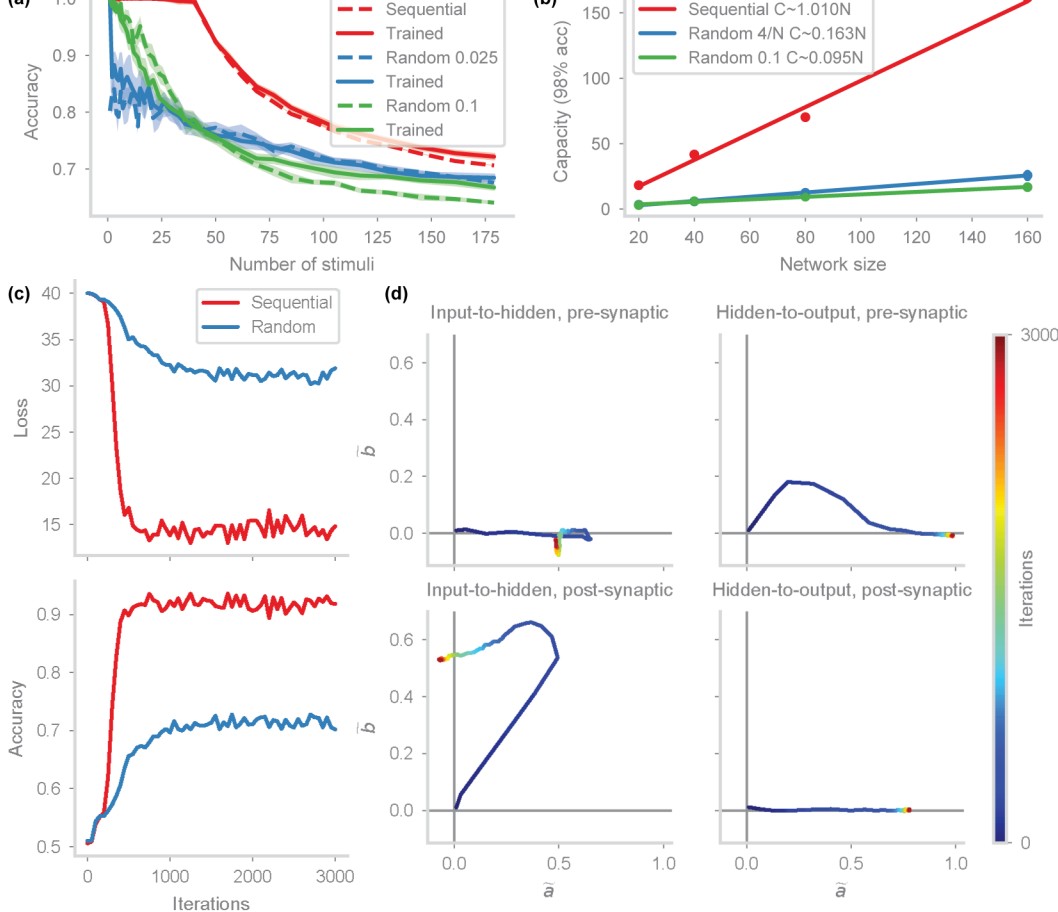

Figure 3: (a) Performance comparison of simple (dashed curves) and meta-learned (solid) network ($d = N = 40$, $p$ varied, shown in legend.) (b) Capacity of trained network with sequential or random local third factor. In the random case, $p$ is either fixed or scales with $N$. (c) Training the sequential and random ($p = 0.1$) network from (a,b). Although accuracy is not $1.0$ during training, the network successfully generalizes to unseen sequence lengths. (d) Learning trajectory of the plasticity parameters in the sequential network converges to qualitatively similar solutions as the original simplified network (see also Figure C4). Note that the input-to-hidden plasticity is independent of the post-synaptic firing rate (bottom left plot, $\widetilde{a}^{g^\kappa} \approx 0$)

consists of sequence lengths between $T = N/2$ and $T = 2N$ (above capacity for both versions of the network), so accuracy does not reach $1.0$. Nevertheless, the network successfully generalizes to lengths outside of this range, indicating a robust mechanism for memory storage.

Most importantly, we examine the plasticity rule discovered by optimization. Figure 3d shows the slope $\widetilde{a}$ and offset $\widetilde{b}$ parameters for the firing rate transfer functions $g^\kappa, f^\kappa, g^\nu, f^\nu$ in the sequential algorithm over the course of meta-learning. The plasticity rule for the input-to-hidden connections (Figure 3d, left) becomes pre-dependent ($\widetilde{a}^{f^\kappa} \approx 0.5$, $\widetilde{b}^{f^\kappa} \approx 0$) but not post-dependent ($\widetilde{a}^{g^\kappa} \approx 0$, $\widetilde{b}^{g^\kappa} \approx 0.5$), analogous to the idealized plasticity rule described in section 2. Similarly, the plasticity rule for the hidden-to-output connections becomes Hebbian (both pre-dependent and post-dependent, $\widetilde{b}^{g^\nu} \approx \widetilde{b}^{f^\nu} \approx 0$, but $\widetilde{a}^{g^\nu} \approx \widetilde{a}^{f^\nu} \approx 1$). The random version shows qualitatively similar trained behavior (Figure C4).

Since the performance and parameterization of the meta-learned algorithm is almost identical to the simple case when using sequential local third factors, we only consider the random version for meta-learning in subsequent sections.

## 3.3 Continual, flashbulb, and correlated memory tasks

We next test our plasticity rules on more ecologically relevant memory tasks. These tasks reflect the complexity of biological stimuli and the functionality needed for versatile memory storage.

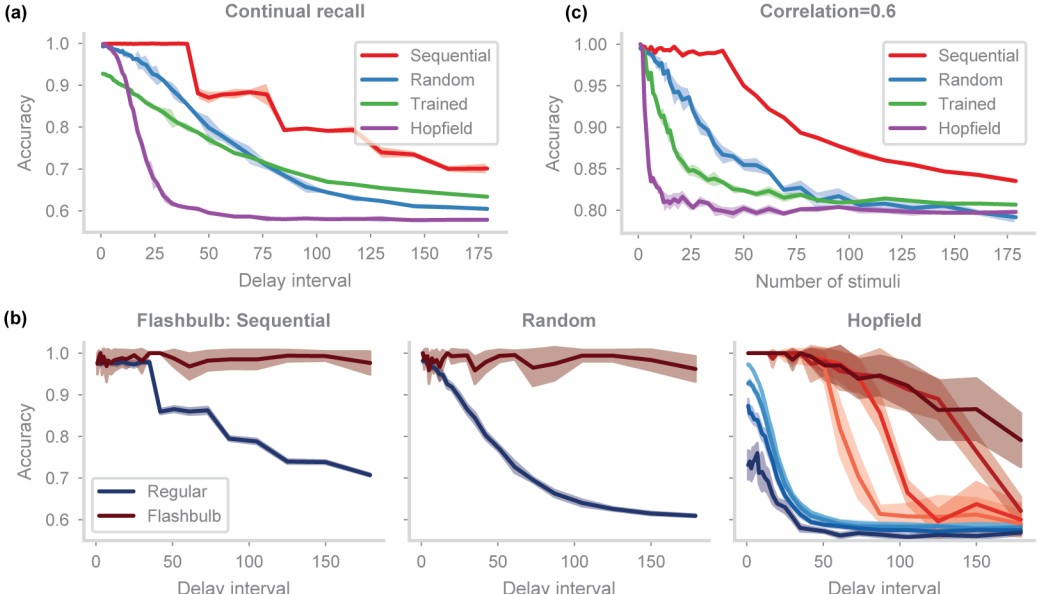

Figure 4: (a) Performance on continual recall task, accuracy measured as a function of the number of timesteps between the storage and recall of a memory. "Trained" corresponds to the meta-learned algorithm with random local third factors. (b) Same as (a), but including five "flasbulb" memories. Performance of simplified algorithm with meta-plasticity, using sequential (left) or random (middle) local third factors. For Hopfield network (right), increasing darkness of each line corresponds to higher write strength for flashbulb memories, with values 10, 50, $10^3$, and $10^6$. (c) Same as Figure 2a, but with memories having correlation of 0.6.

In realistic scenarios, memories are stored and recalled in a continual manner – the subject sees an ongoing stream of inputs and is required to recall a stimulus that was shown some time ago (Appendix A) – rather than being presented a full dataset to memorize before testing, as in the benchmark autoassociative task. Figure A2a shows such a sample dataset with a delay interval of 2. Our learning algorithm is naturally suited for this task due to its decay mechanisms – recent stimuli are stored, but older ones are forgotten. Its accuracy decreases in steps of width $N$ due to the sequential nature of the local third factor (Appendix A). Performance with random third factors decays smoothly since writing is stochastic, but both networks show better overall performance than the Hopfield network (Figure 4a).[4]

Next, we consider "flashbulb" memories, a phenomenon where highly salient experiences are remembered with extreme efficacy, often for life [Brown and Kulik, 1977]. Functionally, these memories may be used to avoid adverse experiences or seek out rewarding ones. We modify the continual recall task such that a small number of stimuli are deemed salient and accompanied by a stronger global third factor ($q_t = 10$) akin to a boosted neuromodulatory influence on learning (Figure A2b). We add a simple meta-plasticity mechanism to our plasticity rule, a stability parameter $[\boldsymbol{S}_t]_{ij}$ for each synapse, initially set to 0. If the learning rate for that synapse crosses a threshold $[\boldsymbol{\eta}_t]_{ij} > 1$, then $[\boldsymbol{S}_{t+k}]_{ij} = 1$ for all $k > 0$ and suppresses subsequent plasticity events. Thus, the learning rate for the key matrix is as follows (learning rate $\boldsymbol{\eta}_t^{\text{v}}$ is analogous):

$$[\boldsymbol{\eta}_t^{\text{K}}]_{ij} = (1 - [\boldsymbol{S}_t^{\text{K}}]_{ij})q_t[\boldsymbol{\gamma}_t]_i \tag{17}$$

---

[4]For fair comparison, we introduce an empirically chosen synaptic decay parameter $\lambda = 0.95$ to the Hopfield network. We also considered a version of the Hopfield network with bounded weights [Parisi, 1986], designed for continual learning, but found that synaptic weight decay has better overall performance. Furthermore, we include a global third factor which controls overall plasticity identically to our model.

With this meta-plasticity, the network retains flashbulb memories with minimal effect on its recall performance on regular memory items (Figure 4b, left, middle). To perform this task in the Hopfield network, introducing synaptic stability is significantly detrimental to performance, so we simply store flashbulb memories as large-magnitude updates to the weight matrix. As a result, it exhibits a tradeoff between storage of regular items and efficacy of flashbulb memories (Figure 4b, right). Thus, an important benefit of the slot-based storage scheme is that flexible treatment of individual memories can be naturally implemented.

Finally, real-world stimuli are often spatially and temporally correlated – for instance, two adjacent video frames are almost identical. Storing such patterns in a Hopfield network causes interference between attractors in the energy function, decreasing its capacity. In contrast, by storing stimuli in distinct slots of the key and value matrices, key-value memory can more easily differentiate between correlated but distinct memories. To verify this, we use a correlated dataset by starting with a template vector and generating stimuli by randomly flipping some fraction of its entries (Figure A2c). The performance of the plasticity rule is similar to that for uncorrelated data (Figure 2a), with minor degradation due to spurious recall of similar stored memories (Figure 4c). Figure C5 shows similar results for varying correlation strengths.

### 3.4    Heteroassociative and sequence memory

The network and learning mechanism is agnostic to the relationship between the input and target patterns, and so naturally generalizes to heteroassociative memory. To draw comparisons with Hopfield-type networks, we consider the Bidirectional Associative Memory (BAM) [Kosko, 1988], a generalization of the Hopfield network designed for heteroassociative recall. We evaluate the networks on a modified version of the recall task from subsection 3.1 where values are distinct from keys, $y_t \in \{+1, -1\}^m$ for $d \neq m$ (Figure 5a, top; Figure A3). Compared to the autoassociative task (Figure 2), the Hopfield-type network's performance on the heteroassociative task deteriorates (Figure 5, bottom). On the other hand, the performance of our network remains unaffected in the heteroassociative task, as its factorized key-value structure allows more flexibility in the types of memory associations learned.

A more biologically relevant version of heteroassociative memory is sequence learning. Experimental and theoretical evidence suggests that hippocampal memory can serve as a substrate for planning and decision making through the replay of sequential experiences [Foster and Wilson, 2006, Pfeiffer and Foster, 2013, Mattar and Daw, 2018]. As a simple probe for a similar functionality, we use a sequence recall task where the network is presented with a sequence of patterns to memorize, using the value at time $t$ as the key at time $t + 1$. Afterwards, it is prompted with a random pattern from the sequence and tasked with recalling the rest. By adding a recurrent loop, using the output $\widetilde{y}_t$ at time $t$ as the input $x_{t+1}$ at the next timestep (Figure 5b, top), our network with sequential local third factors can perform sequence learning without error until its capacity limit (Figure 5b, bottom). BAM does not perform as well, likely due to propagating errors from incorrect recall of earlier patterns in the sequence.

Finally, we consider a more complex task that requires our memory module to be integrated within a larger system, demonstrating the modular nature of our memory network. In the "copy-paste" task [Graves et al., 2014], the system must store a variable-length sequence of patterns $((s_1, s_2, \ldots, s_{T+1}))$ and output this sequence when the end-of-sequence marker is seen (Figure A3). We train an external network as a "controller" to generate $x_t$, $y_t$, and $q_t$ (Figure 5c, top) (for BAM, $q_t$ scales the magnitude of the Hebbian update). With this controller, our network with sequential local third factors successfully learns the task and generalizes outside the sequence lengths seen (Figure 5c, bottom). The random and trained networks do not perform as well, likely due to lower memory capacity. BAM successfully learns the task, but is not able to generalize outside the sequence lengths seen in training.

## 4    Discussion

We proposed models of biological memory by taking inspiration from key-value memory networks used in machine learning. These biologically plausible models use a combination of Hebbian and non-Hebbian three-factor plasticity rules to approximate key-value memory networks. Due to the flexibility of their structure, they can naturally be adapted for biologically relevant tasks. Our

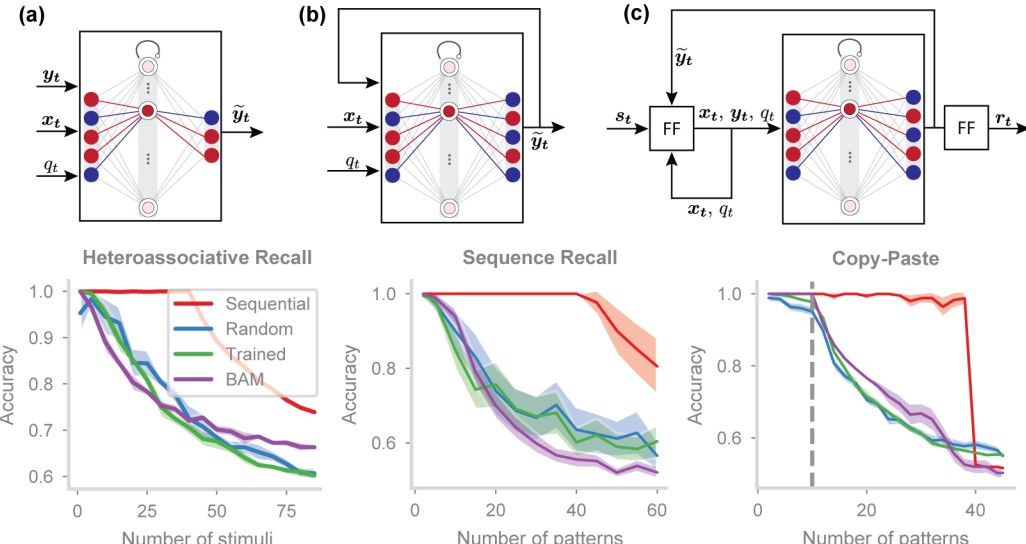

Figure 5: (a) Architecture ($d = N = 40, m = 20$) and performance on heteroassociative version of the benchmark task (Figure 2). (b) Recurrent architecture modification for sequence recall ($d = N = 40, m = 40$), $\boldsymbol{x}_{t+1} = \widetilde{\boldsymbol{y}}_t$. Accuracy is plotted as a function of the length of the entire sequence presented. (c) Network is embedded in a larger system with a feedforward network to perform the copy-paste task ($d = N = 40, m = 40$). Dashed line is the maximum number of patterns shown during training.

results suggest an alternative framework of biological long-term memory that focuses on feedforward computation, as opposed to recurrent attractor dynamics. Importantly, both our hand-designed and meta-learned results propose a role for recently discovered non-classical plasticity rules [Bittner et al., 2017] that are advantageous for this type of feedforward computation. Furthermore, we propose an architecture where individual memories are stored more independently of each other than in Hopfield networks. Such a factorized design creates opportunities for more versatile use and control of memory.

Several questions remain. First, although long-term memory describes many categories of memory supported by various brain regions, we have not disambiguated these differences and their implications on testing and interpreting our model. To validate our network as not merely a plausible model but as a true model of the brain, it is critical to make direct comparisons between our algorithm and experimental findings in memory-related behavior and neural activity. Furthermore, our aim is not to be competitive with state-of-the-art memory networks but rather to provide a biologically realistic learning algorithm for MHNs that is on par with classical Hopfield networks. To this end, we focus on artificial stimuli with simple statistical structures; it remains unclear how our models will perform with more complex and naturalistic data, or how they compare to their non-biological counterparts.

Taken together, our results take a neuroscience-minded approach to connect two lines of work in memory networks that have been largely disparate.

## 5    Acknowledgements

We are particularly grateful for the mentorship of Larry Abbott. We also thank Stefano Fusi, James Whittington, Emily Mackevicius, and Dmitriy Aronov for helpful discussions. Thanks to David Clark for bringing Bidirectional Associative Memory to our attention. Research supported by NSF NeuroNex Award DBI-1707398, the Gatsby Charitable Foundation, and the Simons Collaboration for the Global Brain. G.R.Y. was additionally supported as a Simons Foundation Junior Fellow. C.F. was additionally supported by the NSF Graduate Research Fellowship.

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
