# A  Task details

## A.1  Benchmark: autoassociative recall

For the autoassociative memory benchmark task, we generate $T$ memories, each of which is a uniformly randomly generated $d$-dimensional vector $\boldsymbol{x}_t \in \{+1, -1\}^d$, for $t = 1, \ldots, T$. During the storage phase, the key and value matrices are initialized to zero and each pattern is shown to the network sequentially with the network's global third factor $q_t = 1$ active for all patterns. For storage, both the network's input and output layers are clamped to the input value $\boldsymbol{x}_t$. Next, during the test phase, the network is shown queries $\widetilde{\boldsymbol{x}}_t$, corresponding to the previously shown stimuli with 60% of the entries in each vector randomly set to zero (Figure A1). Queries are shown in the same order as the stimuli and the global third factor $q_t = 0$ to ensure no plasticity occurs. Only the input layer is clamped and the result is read out from the output layer $\widetilde{\boldsymbol{y}}_t$. Accuracy is computed as the total fraction of correctly recalled entries, calculated for varying values of $T$:

$$\text{accuracy} = \frac{1}{Td} \sum_{t=1}^{T} \sum_{i=1}^{d} \mathbb{I}\{[\boldsymbol{x}_t]_i = \text{sign}([\widetilde{\boldsymbol{y}}_t]_i)\} \tag{18}$$

Note that for the classical Hopfield network, the input and readout neurons are the same, so by presenting a query $\widetilde{\boldsymbol{x}}_t$, a fraction of the output bits in $\widetilde{\boldsymbol{y}}_t$ are *a priori* set to the correct values from $\boldsymbol{x}_t$. This raises the chance level of the classical Hopfield network compared to the other networks we consider in Figure 2a.

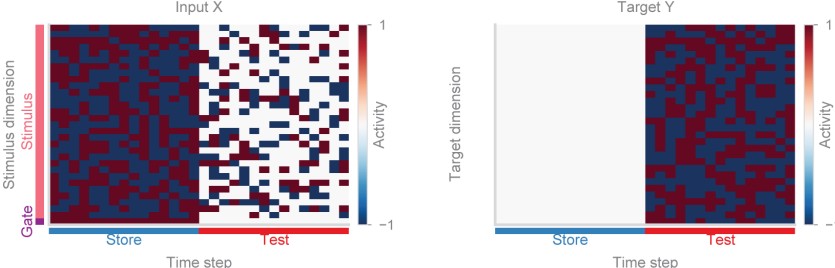

Figure A1: Autoassociative recall benchmark task. $d = 30, T = 15, 60\%$ occluded during test.

## A.2  Beyond simple recall

To test the network in a continual setting, rather than datasets of fixed length $T$, we use arbitrarily long datasets where the network is asked to recall a stimulus that was presented $R$ timesteps ago. To generate the dataset, at each timestep with probability $p_{\text{gen}} = 0.5$ the input $\boldsymbol{x}_t$ is a randomly generated binary vector (as in the benchmark dataset). With probability $1 - p_{\text{gen}} = 0.5$, the input is a query (as in the benchmark dataset) $\widetilde{\boldsymbol{x}}_t$, corresponding to the input shown $R$ timesteps ago[5], $\boldsymbol{x}_{t-R}$. For the generated stimuli which are subsequently queried, the modulator $q_t = 1$ during their initial presentation. Otherwise, $q_t = 0$. To ensure that the network is operating in steady state and therefore in the continual learning regime, we use a long trial duration $T = \max(1000, 20R)$. Figure A2a shows 30 timesteps of such a trial with $R = 2$. Note that in a single trial, the delay interval between the stored stimulus and the query is always a fixed value $R$. However, we test the network on multiple trials, each with a different value of $R$.

Performance of the network with sequential local third factors decreases in steps of width $N$ because it selects the next slot at each timestep regardless of whether the current one was written to. Since a global third factor does not occur at every timestep, some slots left untouched when the local third factor selects them for a second time, thus preserving their contents with a probability which depends on the frequency of queries in the stream. This probability is the same for all delays of length $N + 1$ to $2N$, slightly lower for all delays of length $2N + 1$ to $3N$ (i.e. the slot doesn't get written when the local factor selects it the second *and* the third time), and so forth, resulting in a stepwise curve.

---

[5]We furthermore ensure that a query is not presented twice, so if the input $R$ timesteps ago was a query, a stimulus vector is generated.

The "flashbulb" memory task is similar to the continual task, however for every trial, 5 memories are selected as flashbulb memories. These are generated as the others, but are accompanied by a very strong modulatory input $q_t = 10$ rather than the normal $q_t = 1$. Figure A2b shows a portion of the continual stream, including two of the flashbulb memories.

To test the network performance on datasets with correlated stimuli, we generate $T$ binary random vectors and evaluate the network as in the benchmark task (Figure A1). The first "template" vector is generated randomly $\boldsymbol{x}_1 \in \{+1, -1\}^d$ as before. All subsequent stimuli are generated by randomly flipping a fraction $(1 - \rho)$ of the entries in the template vector, resulting in correlated stimuli with $\text{corr}(\boldsymbol{x}_t, \boldsymbol{x}_{t'}) = \rho$ (Figure A2c). Figure C5 shows the network performance for additional values of $\rho$.

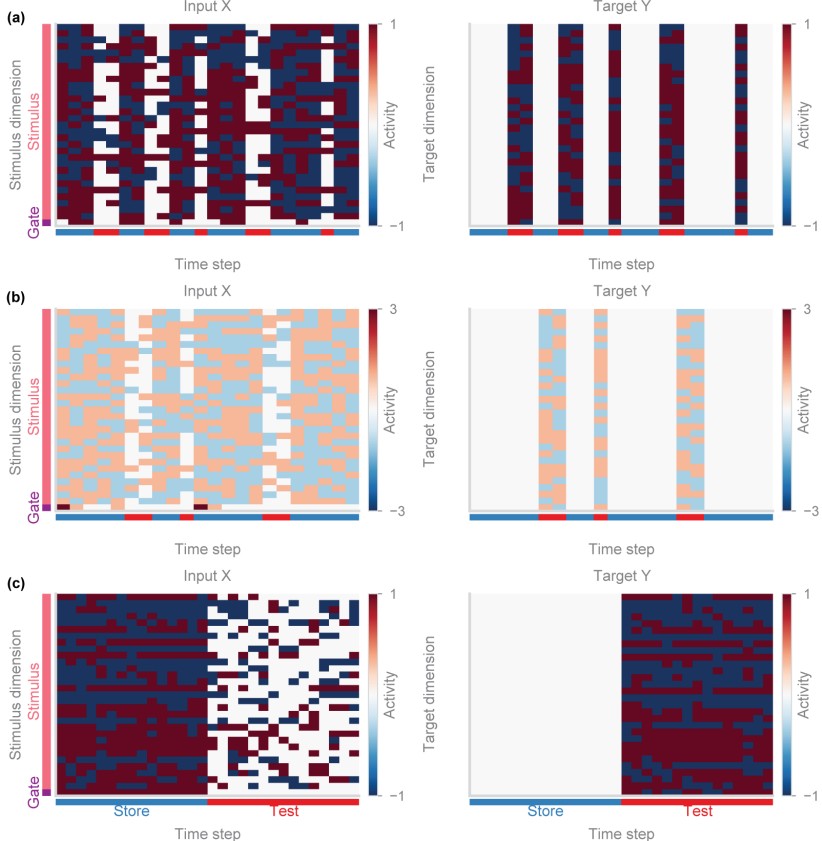

Figure A2: (a) Thirty timesteps of a continual autoassociative recall task with $R = 2$. (b) Thirty timesteps of a flashbulb task, showing two of the flashbulb memories. Note colorbar range. For visualization, modulation strength $q_t = 3$ during flashbulb memories. (c) Autoassociative recall task for correlated memories with $\rho = 0.6$.

## A.3 Beyond autoassociative memory

The heteroassociative recall task (Figure A2a) is identical to the autoassociative memory benchmark task (Figure A1) except we have the additional generation of $m$-dimensional vectors $\boldsymbol{y}_t \in \{+1, -1\}^d$, for $t = 1, \dots, T$. For our task, $m = \frac{d}{2}$. Thus, although during the storage phase the network's input is clamped to some input value $\boldsymbol{x}_t$ as in the autoassociative benchmark, the output layer is clamped to the output value $\boldsymbol{y}_t$.

In the sequence recall task (Figure A2b), similar to the benchmark task, we randomly generate $T$ memories, each of $d$-dimensions. This forms a $T$-length sequence. During the testing phase, a prompt pattern $\boldsymbol{x}_t$ from the middle of this sequence is shown. The goal of the task is to then return the rest of the patterns in this sequence in order: $(\boldsymbol{x}_{t+1}, \boldsymbol{x}_{t+2}, \dots, \boldsymbol{x}_T)$. We run our network recurrently so that, at time $t$, we clamp the network input to $\text{sign}(\widetilde{\boldsymbol{y}}_{t-1})$ and the network output to $\boldsymbol{x}_t$.

In the copy-paste task ([Figure A2]c) we begin by randomly generating a $T$-length sequence as in the sequence recall task. Each pattern $\boldsymbol{s_t}$ is of dimension $D = 25$ (we use a different variable name since the stored keys $\boldsymbol{x_t}$ will be different than the elements of the sequence). We add an additional dimension to each pattern and an additional pattern to the sequence, such that the sequence is $(D + 1) \times (T + 1)$. The additional dimension is used to denote the end-of-sequence (EOS) marker and is set to $-1$ when it is not in use. The EOS marker is shown at the end of the sequence, at time $T + 1$. Thus, the vector shown to the network at time $T + 1$ is $\boldsymbol{s_{T+1}} = [-1 \ -1 \ldots \ +1]$. After seeing the EOS marker, the goal of the task is to repeat the entire sequence $(\boldsymbol{s_1}, \boldsymbol{s_2}, \ldots, \boldsymbol{s_{T+1}})$. During training and evaluation, $T$ is randomly drawn from 1 to 10 in the task.

We use three feedforward controller networks coupled with our memory network. At time $t$, each controller network receives a $(D + 2d + 1)$-dimensional input $\boldsymbol{v_t}$: a concatenation of $\boldsymbol{s_t}, \boldsymbol{x_{t-1}}, \widetilde{\boldsymbol{y}}_{t-1}$ and $q_{t-1}$. The outputs for the three networks are the $d$-dimensional key $\boldsymbol{x_t}$ ($d = 40$), $d$-dimensional value $\boldsymbol{y_t}$, and scalar global third factor $q_t$ for the memory module as follows. Then,

$$\boldsymbol{x'_t} = \tanh(\boldsymbol{R_x} \boldsymbol{v_t} + \boldsymbol{b_x})$$
$$\boldsymbol{y'_t} = \tanh(\boldsymbol{R_y} \boldsymbol{v_t} + \boldsymbol{b_y})$$
$$q_t = \sigma(\boldsymbol{R_q} \boldsymbol{v_t} + \boldsymbol{b_q})$$

where $\sigma(\cdot)$ is the logistic function, and $\boldsymbol{R_x}, \boldsymbol{R_y}, \boldsymbol{R_q}, \boldsymbol{b_x}, \boldsymbol{b_y}, \boldsymbol{b_q}$ are learned matrices. These outputs are normalized to have L2-norm $\sqrt{d}$ to match the norm of the inputs presented in the autoassociative memory task:

$$\boldsymbol{x_t} = \sqrt{d} \frac{\boldsymbol{x'_t}}{||\boldsymbol{x'_t}||} \tag{19}$$

$$\boldsymbol{y_t} = \sqrt{d} \frac{\boldsymbol{y'_t}}{||\boldsymbol{y'_t}||} \tag{20}$$

The controller outputs $\boldsymbol{x_t}, \boldsymbol{y_t}, q_t$ are presented to the memory network, which is updated according to our proposed plasticity rules. With the updated key and value matrices, we retrieve the output of the memory module $\widetilde{\boldsymbol{y}}_t$, using $\boldsymbol{x_t}$ as the query. Finally, $\widetilde{\boldsymbol{y}}_t$ is fed into a one-layer network to transform the output from $d$ dimensions to a $D$-dimensional output $\boldsymbol{r_t}$,

$$\boldsymbol{r_t} = \tanh(\boldsymbol{R_o} \widetilde{\boldsymbol{y}}_t + \boldsymbol{b_o}) \tag{21}$$

where $\boldsymbol{R_o}$ is learned. The values $\boldsymbol{x_t}, \widetilde{\boldsymbol{y}}_t, q_t$ are then fed back into the controller as the input for the next time step, along with $\boldsymbol{s_{t+1}}$. The initial inputs $\boldsymbol{x_0}, \widetilde{\boldsymbol{y}}_0$ and $q_0$ corresponding to $\boldsymbol{s_1}$ are also learned.

When the network is prompted with the EOS marker, the output $(\boldsymbol{r_{T+2}}, \ldots, \boldsymbol{r_{2T+2}})$ should be equal to the original sequence $(\boldsymbol{s_1}, \ldots, \boldsymbol{s_{T+1}})$. We train the network end-to-end with backpropagation through time to minimize mean squared error loss.

For our simulations with BAM, we follow the same controller set-up as above. As is the case for the previous heteroassociative tasks, we use the typical BAM update and update the weight matrix by $\eta \boldsymbol{x_t} \boldsymbol{y_t}^{\mathsf{T}}$ with learning rate $\eta$. The learning rate is modulated by the global third factor, as is the case for our models. However, for training purposes, we found it helpful to scale the learning rate such that $\eta = \frac{1}{40} q_t$.

# B  TVT key-value memory mechanism

TVT is an algorithm that enhances the learning of memory-based agents by combining attentional memory access with reinforcement learning [Hung et al., 2019]. Here, we used the key-value memory mechanism used by the model where inputs were written to memory and attentional memory access was used to read stored inputs from memory. Unlike in the original work, there is no LSTM controller or reinforcement learning component. We simply use the read and write functions to a memory matrix as in the original paper, but do not use the TVT algorithm itself or any of the additional architecture used in the original authors' work.

First, a memory matrix is initialized whose rows will each store one stimulus along with its read strength. There is a reader network and a writer network for the read and write operations respectively.

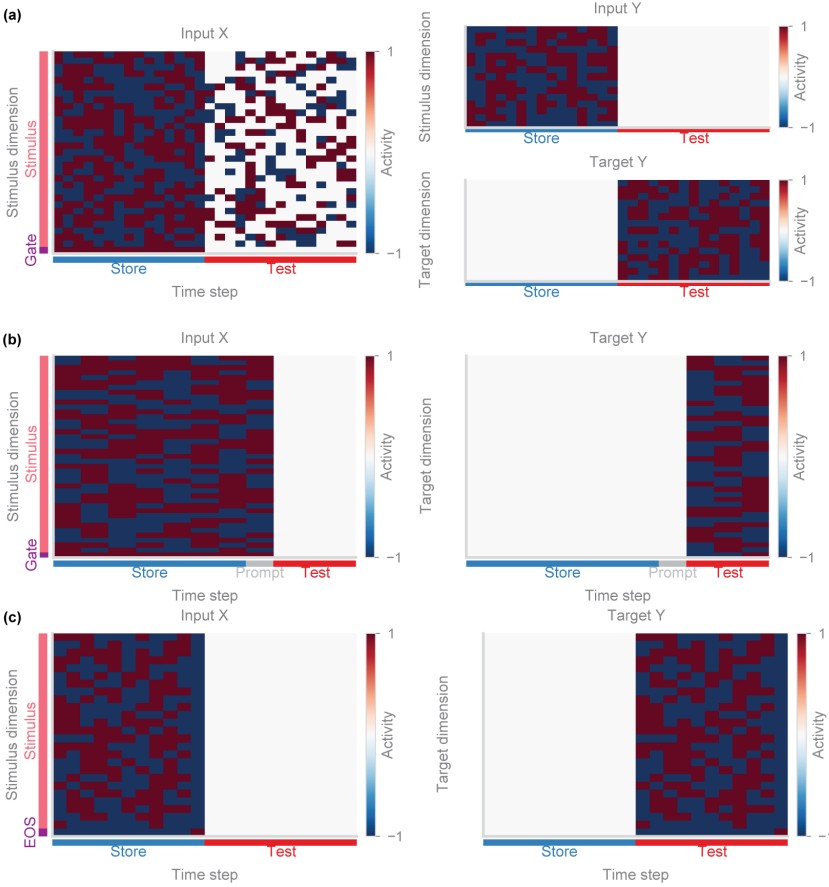

Figure A3: (a) Heteroassociative recall task. $d = 30, m = 15, T = 15, 60\%$ occluded during test. (b) Sequence recall task with $d = 40$ and $T = 7$. The prompt is the 4th pattern of the sequence. (c) Copy-paste task with $d = 8, T = 10$.

A call to write stores a stimulus. A call to read returns the $H$ most similar memories, where $H$ is the number of read heads, or locations that can be read from simultaneously.

For the recall task, the writer receives an index indicating which row in memory should be written to. During a write to memory, the specified row of the memory matrix is cleared and the input vector is written to this cleared slot. During the storage phase, input vectors are stored sequentially such that each incoming input vector is written to the next unfilled row of the memory matrix. If the memory matrix is full, a filled row beginning with the first row will be cleared and an incoming input will be stored in it.

During the retrieval phase of the recall task, the reader uses attentional memory access to retrieve a weighted version the $H$ most similar (smallest in cosine distance) memories from the memory, using $H$ read heads. First, it is given $M \times H$ tensor of inputs where $M$ is the length of each input vector for each of the $H$ read heads. Next, the read keys and the weights used for each key are computed by passing the input through a linear layer that produces an $(M + 1) \times H$ output. The softplus function is then applied to the keys and read strengths output by this linear layer. The resulting $(M + 1) \times H$ tensor is separated into a $M \times H$ tensor of read keys and a $H \times 1$ tensor of read strengths for each read head.

The read keys and the values in the memory matrix are then normalized and multiplied together. This yields a tensor of cosine distances between each read key and each item in memory.

This is multiplied by the $H \times 1$ tensor of read strengths, yielding a $H \times R$ tensor of weighted distances, where $R$ is the number of rows in the memory matrix. These weighted distances are then

passed through a softmax function. Each row then has one element (corresponding to one row in the memory) that is maximally activated. This tensor is then multiplied by the memory matrix, yielding a tensor of memory reads that is a weighted sum of the rows most similar to the weighted read keys.

The linear layer used to generate the keys and read strengths is learned using SGD. A model trained on 20000 steps was used, and with a 40-row memory matrix (to be compared with a size 40 hidden layer of our network).

In the simplified model, unlike in the original paper, only a single read head was used in order to make comparisons with our network. The TVT's key-value memory mechanism works very similarly to the our network with sequential local third factors. The sequential network, however, adds in the biological feature of plasticity rules to store memories.

## C   Supplementary results

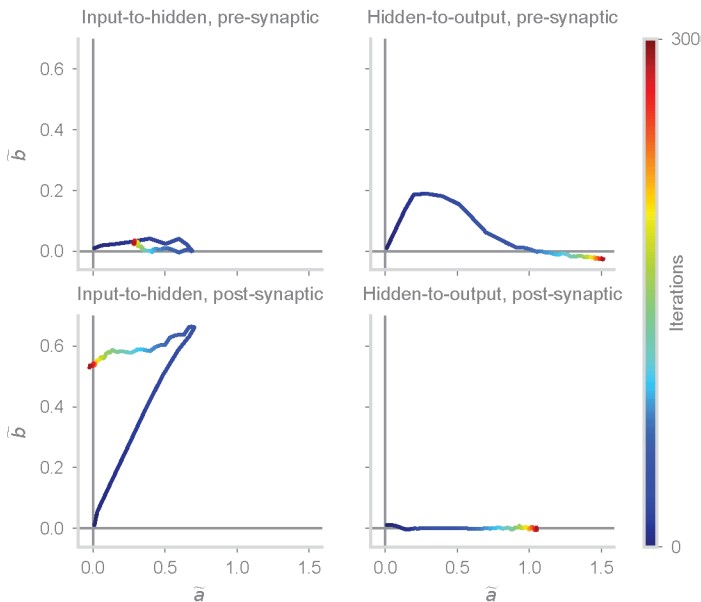

Figure C4: Same as Figure 3d, but for a network with random local third factors.

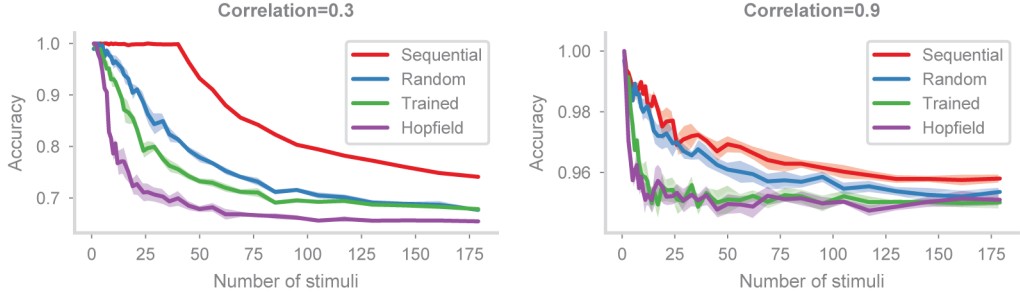

Figure C5: Same as Figure 4c, with different correlation strengths.