# OpenReview forum: "Biological learning in key-value memory networks"
_NeurIPS.cc/2021/Conference — NeurIPS 2021 Poster_

### Official Review · Reviewer_Dwry · 2021-06-25

**Rating:** 6
**Confidence:** 3

**Summary:**

The authors present "Biological Key-value Memory Networks", a biologically plausible method of storing patterns of data in memory for later recall. The method is a simple key-value implementation with each neuron in the hidden layer encoding one key for one memory. The authors provide evidence that this algorithm performs on par with Temporal Value Transport (TVT) and better than Hopfield nets.

_____

Post rebuttal update:
Thanks to the added clarity from the author responses I have increased my score from 4 to 6.

**Limitations And Societal Impact:**

Yes.

**Main Review:**

The contribution is interesting and the paper is generally well written. The authors explore a number of relevant experiments and highlight the strengths of their algorithm. However, because of a number of issues I cannot recommend the work for publication in its current state. The following is a number of strengths and weaknesses of the paper in no particular order.

## Strengths

* The algorithm is intuitive and easy to understand. Furthermore, the biological implementation and learning rules are more or less reasonable.
* The experiments are well varied and thorough in scope. It is also interesting that this simple algorithm tackles tasks that are alternatively solved by more complicated methods (e.g. sequence memorization in bird songs is thought to be handled via RL). Furthermore, the authors show the modularity of their algorithm by building it into other frameworks to perform more complicated tasks.

## Weaknesses

* There is no theoretical analysis of the capacity of the algorithm.
* There are no error bars or confidence intervals in the experiments. Especially for a fast algorithm like this, it won't take much time to run each experiment multiple times and and add confidence intervals. This is particularly necessary for example in Fig. 5, where the accuracy of the network at 60 patterns is higher than at 50, indicating a possibly high error range for the plot. Another example is Fig. 4a, is the plateaud drop in the sequential learning task a real feature or is it just a quirk of that single run of the expriment?
* Comparison to some other models is lacking, e.g. the authors do not mention reservoir networks. Some references to prior work is lacking, e.g. Fixing the weights connected to specific neurons to reduce catastrophic forgetting was previously considered in _Continual Learning via Neural Pruning_ circa 2019.
* Some confusion regarding Hebbian learning rule. In my understanding, Hebbian 'fire together, wire together' implies that the pre and post part of the plasticity are the activities of the neurons. If, as in eq. 9, the learning rule is dependent on a third factor, this is no longer Hebbian.
* In Eq. 9, why not just use the activity $h'_t$ instead of this local third factor? This is what the meta learning rule learns and is motivated already in lines 135-136.
* The simplicity of the algorithm is a little deceiving since it requires a mechanism to indicate when to store and retrieve information. This should be highlighted since competing methods can store and retrieve without queuing (albeit it comes at a cost of performance). In other words, whether or not to store an incoming pattern (information regarding whether it is new or not), needs to be provided to the network, which is why it performs well in the correlated recall task. In a Hopfield net, the dynamics of the evolving energy landscape itself decides if a new energy minimum should be created or if a preexisting minimum should be updated (that is the network itself decides if the information is new and should be stored).  _This needs to be properly addressed in the paper, so the reader understands where the performance improvements are coming from._

## Clarity
Even though the paper is in general readable, there are a number of issues that need to be addressed.
* There are many references to the appendix which are mislabeled (e.g. line 273 A4->A3, line 248 A3->C5.
* Eq. 13 what is $\tilde\eta^y$?
* (optional) I would suggest moving the learning trajectory plots (Fig 3d) to the appendix, and instead using the freed up space for perhaps parts of Fig A1 or A2, which explain the workings of the experiments. While the trajectory is interesting, the main takeaway is that it started from zero and ended at the reported values in lines 209-211.

**Time Spent Reviewing:**

5

---

> ### Author Response · Authors · 2021-08-10
> **Addressing weaknesses and providing clarifications**
>
> We thank the reviewer for their careful analysis of our work, and believe their suggestions will significantly improve the quality of our submission. We have addressed each potential weakness below by proposing revisions or providing clarifications.
>
> - There is no theoretical analysis of the capacity of the algorithm.
>   - In the sequential case, the capacity is, exactly equal to the number of neurons since each neuron corresponds to a memory slot, and overwriting only occurs after every slot has been filled. The random case is more difficult, and we will make an effort to include this calculation.
> - There are no error bars or confidence intervals in the experiments. Especially for a fast algorithm like this, it won't take much time to run each experiment multiple times and and add confidence intervals. This is particularly necessary for example in Fig. 5, where the accuracy of the network at 60 patterns is higher than at 50, indicating a possibly high error range for the plot. Another example is Fig. 4a, is the plateaud drop in the sequential learning task a real feature or is it just a quirk of that single run of the expriment?
>   - To address this issue, we will add error bars for all plots, and make the two additional clarifications discussed below.
>   - We have observed very small variance for almost all of the plots, and almost none in the sequential version of the algorithm. One notable exception, as the reviewer points out, is in Fig. 5 where the network is running a more complex task, and in conjunction with a second “controller” network. As a result of the additional task and architectural complexity, the variance is higher.
>   - In Fig 4a, the plateaus are indeed a real feature due to limitations of the network size. If N=50, any delay intervals between 1 and 50 are equally likely to be recalled since the dataset is below the network’s capacity. If the delay interval is between 51 and 100, there is some error since there are now two memories overlapping in each slot. For delays between 101 and 150, there are three memories per slot, and so forth.
> - Comparison to some other models is lacking, e.g. the authors do not mention reservoir networks. Some references to prior work is lacking, e.g. Fixing the weights connected to specific neurons to reduce catastrophic forgetting was previously considered in Continual Learning via Neural Pruning circa 2019.
>   - Thank you for bringing our attention to these references. Could you please provide an example reference for reservoir networks, or  additional details? To our knowledge, reservoir networks e.g. those considered AS2007, JH2004 or W2007 are not designed to serve as a rapid key-value memory storage system.
>   - Regarding CLNP, please note that our model learns continually out-of-the box without catastrophic forgetting, without any need to fix any weights. In the flashbulb task, we fix weights using the stability factor to extend the lifetime of memories, rather than to enable storage of additional ones (similar to how flashbulb memories, as defined in the psychology literature, are extremely salient and may last a lifetime). We believe this is distinct from CLNP which proposes a solution to catastrophic forgetting, common in networks trained with SGD.
> - Some confusion regarding Hebbian learning rule. In my understanding, Hebbian 'fire together, wire together' implies that the pre and post part of the plasticity are the activities of the neurons. If, as in eq. 9, the learning rule is dependent on a third factor, this is no longer Hebbian.
>   - Thank you for pointing out this potential source of confusion -- this is an issue of language, and we will make the following clarification. Three-factor learning rules have been described as “neo-Hebbian” [G2018]. The more appropriate description in this case would be “neurons that fire together are eligible to wire together,” where that eligibility is then converted to plasticity by a third factor. An alternative view is that the learning rule is the same as a traditional Hebbian rule, but with a variable learning rate (i.e. our prefactor \eta). Generally, we use the term “Hebbian” to differentiate a “pre-times-post” rule (with or without a third factor) from a “pre-only” or “post-only” non-Hebbian rule.
> - In Eq. 9, why not just use the activity h instead of this local third factor? This is what the meta learning rule learns and is motivated already in lines 135-136.
>   - If we are understanding correctly, the reviewer proposes to use $\eta^v_{ij}(t) = q(t)h_i'(t)$ instead of  $\eta^v_{ij}(t) = q(t)\gamma_i'(t)$  in eq. 9. Correspondingly, this can also be used in eq. 13 as a rapid decay mechanism instead of a passive decay via $\lambda_t$. The simplest advantage is that local third factors are truly binary, whereas the hidden activity is only approximately one-hot, so the local third factor has a (minor) performance boost. Using the activity $h’(t)$ as a rapid decay mechanism could indeed confer an advantage over the passive decay in eq. 13 while still maintaining biological realism. We empirically observe that passive decay (with an appropriately tuned decay factor) operates nearly as well as the upper-bound performance of the local-third-factor-based rapid decay. Nevertheless, this is a very interesting suggestion, and we will explore it further.
> - The simplicity of the algorithm is a little deceiving since it requires a mechanism to indicate when to store and retrieve information. This should be highlighted since competing methods can store and retrieve without queuing (albeit it comes at a cost of performance). In other words, whether or not to store an incoming pattern (information regarding whether it is new or not), needs to be provided to the network, which is why it performs well in the correlated recall task. In a Hopfield net, the dynamics of the evolving energy landscape itself decides if a new energy minimum should be created or if a preexisting minimum should be updated (that is the network itself decides if the information is new and should be stored). This needs to be properly addressed in the paper, so the reader understands where the performance improvements are coming from.
>   - The storage/retrieval indicator is the global third factor $q_t$. Indeed, other key-value networks such as the Neural Turing Machine (NTM) or its extensions generate this indicator without manual/external cues. One possible solution to this issue is to generate $q_t$ internally rather than providing it, for instance based on the maximal distance between the query and each stored value. If the distance is below a threshold, $q_t$ should be 1 and 0 otherwise.
> However, note that the only use case for this indicator in our work (with the exception of the flashbulb task) is to delineate the learning and evaluation phases, rather than to modulate the novelty of an incoming stimulus. That is, $q_t=1$ for every input (see “Gate” indicator in the task description figures in the Appendix). For a fair comparison, we control the plasticity Hopfield network in a similar way where plasticity only occurs during the storage phase. During evaluation, both networks are frozen so that probing one stimulus does not affect the others. Thus, our model’s advantage over the Hopfield network is not due to this indicator but rather due to an architectural and/or learning algorithm advantage.
>
> Finally, regarding clarity, thank you for the careful attention to these details. We will correct the references to the appendix. $\eta^y$ should be $\eta^v$ in eq. 13. Moving the learning trajectories to the appendix would indeed use the space more effectively and allows us to make the corrections suggested above.
>
> [AS2007] Abbott LF, Sussillo D, Generating Coherent Patterns of Activity from Chaotic Neural Networks https://doi.org/10.1016/j.neuron.2009.07.018
>
> [JH2004] Jaeger H, Haas H, Harnessing Nonlinearity: Predicting Chaotic Systems and Saving Energy in Wireless Communication 10.1126/science.1091277
>
> [M2007] Maas W, Joshi P, Sontag E, Computational Aspects of Feedback in Neural Circuits https://doi.org/10.1371/journal.pcbi.0020165
>
> [G2018] Gerstner W, et al., Eligibility Traces and Plasticity on Behavioral Time Scales: Experimental Support of NeoHebbian Three-Factor Learning Rules https://doi.org/10.3389/fncir.2018.00053

---

> > ### Comment · Reviewer_Dwry · 2021-08-11
> > **Post-Rebuttal comment**
> >
> > Thank you for the detailed response to the individual questions. I am still a little confused on some points:
> >
> > 1. The authors claim that they do not use the third factor except to delineate the training and evaluation phases. However, is it not true that in the sequential training phase, each incoming input is saved to the 'next' available (or some random) neuron? The fact that you know to save this to the next neuron as opposed to a previously used neuron is the main advantage of the sequential training in my opinion. In this way the third factor is crucial. As I mentioned in my initial review, a Hopfield net, which does not have access to this information, has to dynamically figure out how to modify its energy landscape (i.e. modify a minimum corresponding to a previously trained memory, or create/modify another minimum.)
> >
> >     I agree with the authors that the implementation of a threshold would be a way to equalize the playing field for the different algorithms. I would be very interested to see these results and would be surprised if the sequential training maintains its significant lead over other algorithms when this thresholding method is implemented. As such, I still think that the authors need to *explicitly* mention this difference between their work and prior work (e.g. Hopfield nets). And also provide the readers some insight as how to figure out when a sample is new (similar to the thresholding method mentioned) and ideally provide experiments that probe this method (i.e. how much does using thresholding to figure out if you have a new sample actually cost you in performance.)
> >
> > 2. References.
> >
> >     * An example of neural reservoirs used for memory retrieval is *Rodriguez N, Izquierdo E, Ahn YY. Optimal modularity and memory capacity of neural reservoirs. Netw Neurosci. 2019;3(2):551-566.*
> >
> >    * I am puzzled by the claim that the authors do not fix any weights. The algorithm is wholly based on fixing the weights of all neurons except for the single neuron that is being trained at that time step. And after being trained, its weights are again fixed until the network runs out of memory (or in the random case, until it is randomly drawn for training again). In this sense, this network is not unlike CLNP, where the fixing of the individual weights was determined dynamically, whereas here they are fixed in a predetermined fashion. (Granted CLNP is much more complicated to implement and there authors do not consider biological implementation.)

---

> > > ### Author Response · Authors · 2021-08-12
> > > **Additional clarifications**
> > >
> > > Thank you for your clarifications of your concerns, we hope the following can further explain these points:
> > >
> > > 1. If our understanding is correct, the reviewer’s original concern was regarding the local third factor indicating the location (neuron) for storage, rather than the global third factor indicating whether to store at all as we originally assumed. To this end, we would like to reiterate and emphasise the following points:
> > >  - The global third factor $q_t$ (provided as an external input in our experiments, but may be implemented via a thresholding as discussed) does, in fact, only delineate the training and evaluation phases. It does not give our network an advantage since we provide the same signal to the Hopfield network (for both networks, we do not update the weights if $q_t=0$).
> > >  - The reviewer is correct that the local third factor $[\gamma_t]_i$ (where $i$ indexes the hidden neuron) selects the next/random neuron for storage. Indeed, in the sequential case, knowing which neuron is “next” (i.e. having an internal state) provides the advantage. However, in the random case, no additional information/internal state is used when selecting a neuron for storage so our network and the Hopfield network are on equal ground.
> > >
> > > To clarify, we will add a discussion of these two points, as well as the suggestion for implementing an internal method for generating the global third factor, e.g. via thresholding or a familiarity signal.
> > >
> > > 2. Thank you for providing the Rodriguez et al. reference, it is a very interesting study investigating the effect of modularity on memory capacity and duration in ESNs. The learning and evaluation procedure, however, appears to be in spirit different from ours, requiring task-specific training to generate a set of attractors for memory storage via neuronal dynamics, rather than the one-shot memorization of arbitrary inputs via synapses which we consider in our work.
> > > 3. Thank you for the clarification regarding weight fixing. From this perspective, it makes sense to say we are fixing all weights except those selected by the local third factor. Indeed, sparse updates to weights have many advantages, however we are looking at memory capacity, whereas CLNP is looking at continual learning. There is a connection between the two, but it is beyond the scope to do a thorough comparison with the continual learning literature. In addition, please note that in the meta-learned version of the network none of the hidden-to-output weights are fixed -- not only do they all undergo a global decay but also they are updated according to the hidden layer activity which may not be one-hot, particularly if the stored memories are correlated.

---

> > > > ### Comment · Reviewer_Dwry · 2021-08-13
> > > > **Reviewer Response to Additional clarifications**
> > > >
> > > > Thank you for the clarification. This addresses most of my concerns. I also agree with the authors that a thorough comparison with the mentioned works is not necessary for this paper, and it should be sufficient to only cite the references (and any other relevant references regarding reservoir memory networks and weight fixing) such that the reader can follow up if they so choose.

---

### Official Review · Reviewer_dTBY · 2021-07-09

**Rating:** 5
**Confidence:** 4

**Summary:**

The paper proposes KEVIN--a biologically plausible key-value memory using a 2-layer feed-forward neural network wherein the key and value matrices are the weights of each layer. Here, each row of the key matrix is updated toward an input vector (non-Hebbian update). The value matrix is updated using  Hebbian learning. To control the synaptic plasticity rate, the authors use a global and two alternative local (sequential or random) third factors that take inspiration from the brain. In addition to the main design, there are modifications for specific tasks such as learning to scale synaptic plasticity rates with gradient descent and disabling subsequent weight updates conditioned on the current learning rate. The authors compare the proposed model with Hopfield networks in various synthetic tasks, showing competitive results.

**Limitations And Societal Impact:**

Yes

**Main Review:**

The method is incremental as it combines different known techniques (slot-based and Hebbian writing), although there are interesting connections to biological mechanisms in the brain. One interesting finding is that the simplified local third factor and the trained one show similar behaviors. The model is simple and claimed to be implementable with biological neurons. I am curious about how the Sequential KEVIN is biologically plausible, especially how to implement Eq. 4 with real neurons. Also, the proposed architecture seems related to a classic model ADAM [1] (also 2-layer feed-forward net). The idea is also to map the input to some pattern (index) and use it to retrieve the value. It is good to have ADAM as one baseline in the experiment.

Overall, the paper is easy to read. However, there remain unclear details of implementations. For example, how is $q_t$ (in Eq. 6) computed?  What is the exact loss to train f, g (in Eq. 12, 13)?  Some part of writing is confusing: where are Fig. A5 (Line 213) and A4 (line 256)? Fig. 5 caption, why $y_t = x_{t-1}$?

My main concern is the experiment results. Not to mention the toy tasks, the overall performance of KEVIN is not impressive. The best performer is Sequential KEVIN, yet it is obvious thanks to the sequential slot-based writing and it is unclear this mechanism is biologically plausible. Hopfield network still shows competitive and better performance in Auto-associative Recall task. In other tasks, KEVIN shows improvement. However, this is attributed to additional modifications and specific designs for the task, which makes the model no longer simple and generic. Besides, the authors should consider comparisons with gradient-based baselines such as RNN to validate the usefulness of KEVIN.

[1] J. Austin. ADAM: A distributed associative memory for scene analysis. In M Caudhill and C Butler, editors, First International Conference on Neural Networks, volume IV, page 285, San Diego, 1987.

**Time Spent Reviewing:**

4h

---

> ### Author Response · Authors · 2021-08-10
> **Reframing our work and providing clarifications**
>
> We thank the reviewer for their thoughtful evaluation of our work, and their suggestions. A primary concern of the reviewer is the performance of our model. We suggest the following reframing of our manuscript to address this concern.
>
> Many key-value memory networks have been proposed, including Neural Turing Machines, Differentiable Neural Computers, Temporal Value Transport, etc. More recent Modern Hopfield Networks (MHNs) [KH2016, RA2020, KH2020] further implement a similar key-value network with a biologically-plausible architecture. However, MHNs are trained with biologically unrealistic methods. MHNs are the main connection between our model and previous key-value memory networks. As noted by Reviewer 3w4j, our model KEVIN is architecturally equivalent to MHNs, but we use biologically-plausible synaptic update rules in our model. Thus, our interest is in providing a biological learning mechanism for MHNs, and not in comparing the performance of MHN-like architectures to other memory network architectures. To clarify this point in our paper, we propose editing Section 2 and Section 3 to reframe the main contribution of our work:
> - Prior memory networks such as Neural Turing Machines are biologically inspired, but not biologically implemented.
> - MHNs provide an architectural solution to this problem, but are lacking a biological learning algorithm.
> - The KEVIN network architecture is a special case of the MHN architecture.
> - Our main contribution is the pair of local three-factor learning rules.
> - We do not aim to be competitive with state-of-the-art non-biological memory networks, but rather provide a biologically realistic learning algorithm for MHNs that is competitive with classical Hopfield networks.
>
> The addition of a biological learning rule makes MHNs a viable “plug-and-play” alternative to classical Hopfield networks, often considered to be a model of the hippocampus, and is intended to be used as a component in computational neuroscience models rather than a competitor to non-biological memory nets. Thus, it would not be a fair comparison to make a side-by-side comparison with such networks, nor with gradient-based approaches. Nevertheless, we would like to emphasise that the KEVIN network is mathematically equivalent to a special case of the recent TVT network, and has identical performance as demonstrated in Fig 2a.
>
> Regarding Hopfield net's performance on the auto-associative task, due to its architecture, the Hopfield network’s input neurons are the same as the output neurons. When it receives a query (i.e. a corrupted input), the output is already initialized with a partial solution (the uncorrupted bits are set to the desired output values). Thus if fraction p of the input bits are corrupted, the chance level is p+(1-p)/2. In contrast, the KEVIN network’s chance level is exactly 50% regardless of the input, since the output units do not have any a priori information about the desired output.
>
> Regarding the biological plausibility of Sequential KEVIN: Although the sequential version is intended as an upper-bound on performance rather than a literal interpretation, the sequential writing can be biologically justified in several ways. First, each neuron may have an internal timing mechanism that deploys a local third factor every $N$ timesteps. Alternatively, the neurons may be wired such that a plateau potential in neuron $i$ primes neuron $i+1$ for firing a plateau potential at the next time step. Finally, the local third factors may be controlled by an external circuit that coordinates their firing.

---

### Official Review · Reviewer_vmVW · 2021-07-16

**Rating:** 5
**Confidence:** 3

**Summary:**

This paper proposes a biologically plausible neural circuit KEVIN that approximates key-value memory (slot) networks. The neural circuit is a key-value memory network with a key matrix and a value matrix.
“Pre-only” rules are used to store inputs in the key matrix slots based on presynaptic activity, gated by a “local third factor”: Cycling through slots sequentially or via independent slot-probabilities. This local third factor is also used to allow for overwriting/forgetting of key matrix slots. A scalar “global third factor” is used to (de)activate storage of an input independently of individual slots.
In the simplified version, this local third factor is also used for writing to the value matrix. These update rules are then refined and parameterized to interpolate between “pre-only” and “post-only” rules.
KEVIN is analyzed on and evaluated against classic Hopfield networks in three simple long-term memory tasks (continual storage, storage of single event over long time, storage of correlated inputs) and a heteroassociative memory task (sequence memorization). Optimization of the parameterized rules in meta-learning experiments shows that the rule for key matrix slots becomes “pre-only” and the rule for value matrix becomes both pre- and post-dependent.

**Limitations And Societal Impact:**

The authors did not address the societal impact of their work

**Main Review:**


## Major:
The paper proposes and motivates an interesting biologically more plausible key-value memory network. However, the proper placement of the new method and its discussion w.r.t. other state-of-the-art methods are insufficient. The experimental setup, in this form, is inadequate (no error bars, no natural/more complex tasks).

### Originality
1.  The proposed method with its biologically plausible aspect, to my knowledge, is new.
2.  The state-of-the-art w.r.t. possible storage capacity is not reflected appropriately in the paper. While classic Hopfield networks, as pointed out and compared to in the paper (e.g. Section 4.1), indeed exhibit linear storage capacity, modern Hopfield networks achieve exponential storage capacity [1][2][3][4]. One could argue about the biological plausibility of modern Hopfield networks but regardless, it should be mentioned that KEVIN with its linear storage capacity would scale exponentially worse than modern Hopfield networks. As a result, KEVIN would likely be outperformed by such modern Hopfield networks in the experiments conducted in the paper. W.r.t. the correlation task, the learned mapping of the patterns into the associative space would furthermore allow to separate patterns that are correlated in the input space well in the association space. ([1] and [3] are cited in the Introduction but the relation of the KEVIN storage capacity to this state-of-the-art is missing. This would be relevant for the assessment of KEVIN’s performance in the experimental section and its potential for realistic applications. It would also emphasize the impact of such networks given their recent successes in i.a. [3][4][5].)

### Quality
3.  See point 2 - comparison of KEVIN storage capacity only to classic Hopfield networks is misleading.
4.  The paper keeps pointing out a clear gap between Hopfield networks and key-value memory networks. With the connection of classic Hopfield networks to modern Hopfield networks [3] and thereby key-value memory networks, such as the Transformer [5], this statement is too general and should be refined and made more specific.
5.  The experimental section and plots in the paper lack information about repetition of runs with different random seeds and have no error bars.
6.  Experiments on more realistic/complex datasets (e.g. RNN benchmarking datasets) would have been important to judge the performance of KEVIN in larger systems (see point 7).

### Clarity
6.  More information on the experimental setup in the main paper would be useful (see minor comments).

### Significance:
7.  The biologically more plausible aspect is interesting but the significance is hard to judge without experiments on more realistic datasets, especially with the poor performance on the one more complex copy-paste task.

## Minor:
1.  The experimental setup should be described in more detail in the main paper (e.g. how large the FF Controller and KEVIN network is in Section 4.4.).
2.  Figure 1: Please also mention the variable letters with their names in the caption.
3.  Figure 1: The positioning/illustration of the local third factor for gating is a bit confusing, maybe this could be refined (otherwise it looks like the local third factor is the hidden state w.r.t. Fig1.a and Fig.1.b).
4.  Figure 2: Please specify what “Network size” refers to (e.g. number of hidden neurons).
5.  Figure 2: It would be interesting to also have a plot with the number of synapses instead of the number of neurons on the x-axis.
6.  Figures: Maybe the colors could be modified so that they are distinguishable in grayscale for print and color vision deficiency.
7.  W.r.t. memory capacity of RNNs, [6] would also be relevant in the Introduction.
8.  Since $\mathbf{q}$ is used for the query vector in Eq1, it might be beneficial to rename the global third factor to something other than $q_t$.
9.  Typos: Eq.3: is it not $\mathbf{V}^T_t$?; 104: “slots”;

## References

[1] Krotov, D., & Hopfield, J. J. (2016). Dense associative memory for pattern recognition. Advances in neural information processing systems, 29, 1172-1180.

[2] Demircigil, M., Heusel, J., Löwe, M., Upgang, S., & Vermet, F. (2017). On a model of associative memory with huge storage capacity. Journal of Statistical Physics, 168(2), 288-299.

[3] Ramsauer, H., Schäfl, B., Lehner, J., Seidl, P., Widrich, M., Gruber, L., ... & Hochreiter, S. (2020). Hopfield Networks is All You Need. In International Conference on Learning Representations.

[4] Widrich, M., Schäfl, B., Pavlovic, M., Ramsauer, H., Gruber, L., Holzleitner, M., ... & Klambauer, G. (2020). Modern Hopfield Networks and Attention for Immune Repertoire Classification. In Advances in Neural Information Processing Systems (pp. 18832-18845).

[5] Vaswani, A., Shazeer, N., Parmar, N., Uszkoreit, J., Jones, L., Gomez, A. N., ... & Polosukhin, I. (2017). Attention is all you need. In Advances in neural information processing systems (pp. 5998-6008).

[6] Leibold, C., & Kempter, R. (2006). Memory capacity for sequences in a recurrent network with biological constraints. Neural computation, 18(4), 904-941.


**Time Spent Reviewing:**

20

---

> ### Author Response · Authors · 2021-08-10
> **Reframing the main contribution of our work**
>
> Thank you for your time and detailed review.
>
> A primary concern of the reviewer is the comparison of our model to other state-of-the-art methods on more naturalistic tasks (raised in the “Major” and “Significance” sections of the review). We suggest the following reframing of our manuscript to address these concerns. Many key-value memory networks have been proposed, including Neural Turing Machines, Differentiable Neural Computers, Temporal Value Transport, etc. Another class of memory networks called Modern Hopfield Networks (MHNs) [KH2016, RA2020, KH2020] have been proposed to implement these previous key-value networks with a biologically-plausible architecture. However, MHNs are trained with biologically unrealistic methods. MHNs are the main connection between our model and previous key-value memory networks. Our model KEVIN is architecturally equivalent to MHNs, but we use biologically-plausible synaptic update rules in our model. Thus, our interest is in providing a biological learning mechanism for MHNs, and not in comparing the performance of MHN-like architectures to other memory network architectures. To clarify this point in our paper, we propose editing Section 2 and Section 3 to reframe the main contribution of our work:
> - Prior memory networks such as Neural Turing Machines are biologically inspired, but not biologically implemented.
> - MHNs provide an architectural solution to this problem, but are lacking a biological learning algorithm.
> - The KEVIN network architecture is a special case of the MHN architecture.
> - Our main contribution is the pair of local three-factor learning rules.
> - We do not aim to be competitive with state-of-the-art non-biological memory networks, but rather provide a biologically realistic learning algorithm for MHNs that is competitive with classical Hopfield networks.
>
> We next address the points raised in the “Quality” section of the review:
>
> 3.,4.,6. We hope the reframing of the paper will clarify these statements. Specifically, classic Hopfield networks are in contrast to modern Hopfield networks that implement a key-value approach to memory storage. Our network is a more biological implementation of modern Hopfield networks.
>
> 5. We will include error bars in the updated manuscript.
>
>
> Thank you for the additional feedback on the clarity of details in the paper. We address the “Minor” feedback section here:
> 1. We will include more details of the experimental setup in in the main text.
> 2. We will clarify the variable letters in the caption to Figure 1
> 3. “Figure 1: The positioning/illustration of the local third factor for gating is a bit confusing, maybe this could be refined (otherwise it looks like the local third factor is the hidden state w.r.t. Fig1.a and Fig.1.b).”
> Thank you for pointing this out. We can use a different color to outline the neuron where the local third factor is active (e.g. green outline instead of red outline in Fig 1.a). Please let us know if this would help with clarity.
> 4. In Figure 2, network size does refer to the number of neurons. We will add this into the figure caption.
> 5. “Figure 2: It would be interesting to also have a plot with the number of synapses instead of the number of neurons on the x-axis.”
> Thank you for the suggestion. We had touched upon this in line 170: “..with the same number of hidden neurons, KEVIN has twice as many connections as Hopfield networks.” Thus, if capacity was a function of synapses, the capacity of the networks would be Random < Hopfield < Sequential and the difference in capacity between Hopfield and Sequential would be less dramatic. We can expand on this sentence more in the main text.
> 6. “Figures: Maybe the colors could be modified so that they are distinguishable in grayscale for print and color vision deficiency.”
> Thank you for the thoughtful suggestion. We will look into appropriate color palettes.
> 7. “W.r.t. memory capacity of RNNs, [6] would also be relevant in the Introduction.”
> Thank you for the reference. [6] seems most relevant to the discussion of sequence capacity in Hopfield-type networks (specifically, the Bidirectional Associative Memory we simulate). This can be incorporated into the discussion we have in lines 268-269.
> 8. We will use a different variable for the global third factor in equation 1 to prevent confusion with the query vector.
> 9. Typos: Equations 1 and 3 are inconsistent (V is transposed in one but not the other). We will correct this. Line 104 is indeed a typo and we will change “slot” to “slots”.

---

> > ### Comment · Reviewer_vmVW · 2021-08-25
> > **Post-rebuttal comments**
> >
> > I thank the authors for their detailed response.
> >
> > The reframing suggested by the authors appears clearer to me and would dispel my concerns about the placement of the proposed method. While not mentioned in the author response explicitly, I still believe that it would be helpful for the reader to also mention the theoretically exponential storage capacity of MHN/transformers when analyzing the linear storage capacity of KEVIN (at this intersection of biological plausibility and state-of-the-art performance in ML, it seems appropriate and highly interesting to me to briefly mention what the theoretical capabilities of the state-of-the-art are).
> >
> > Other minor points were addressed by the authors.
> >
> > I am still not convinced about the significance of the experimental results (see point 7 of my review) but since the placement of the reframed paper seems appropriate to me, I updated my score (4->5).

---

### Official Review · Reviewer_dV7b · 2021-07-22

**Rating:** 6
**Confidence:** 5

**Summary:**

The paper proposes a new ML-inspired model for biological associative memory. Auto-associative memory networks are a popular model for biological memory, with one-shot learning implemented by hebbian learning and attractor dynamics / gradient descent for retrieval. In contrast, a traditional key-value memory system has fast feedforward retrieval (linear-nonlinear-linear map), with more complex, often gradient based, parameter optimization. Here the authors develop local learning rules for the parameters of a standard key-value network, linking its properties to some recent experimental observations in rodent hippocampus. They also generalize the basic setup to an online metalearning scenario where the gating of plasticity and the functional form of the synaptic updates are themselves parametrized, with hyperparameters adapted to the learning task. The learning procedures are applied to a range of memory tasks and compared to traditional Hopfield networks.




**Ethical Concerns:**

No concerns.

**Limitations And Societal Impact:**

No obvious negative impact beyond that of standard uses of regular key-value memory systems.

**Main Review:**

Overall: The idea is extremely obvious: copy one-to-one the item to be stored into one unique set of synapses, which results into an essentially one hot encoding of the item; learn the reverse map by correlations (essentially copying the target output into the corresponsing readout weights). The mechanistic interpretation seemed unnecessarily convoluted i would think that if you have a random ultra sparse activation in the hidden layer, this activity can simultaneously gate the learning of both the key(in) and value (out) weights at the same time, without additional gating. Presumably the extra steps are there for biological plausibility, not the core computation. Some of the numerical experiments are set in a nonstandard manner and some of the numerical experiments could be improved (see detailed comments). I appreciate the ambition of exploring such a range of tasks and variants but naturally this comes at the cost of getting less insight from the individual pieces. Sometimes less (done extra thoroughly) is more.

Detailed comments:
- i don't quite understand why the two learning processes need to be different, as long as the softmax is more like a WTA, so the activity in the hidden layer is very sparse h should be able to safely gate
- i have some conceptual issues with benchmark 1: in the traditional hopfield net the cue is a valid pattern - in a spin (+/-1) representation the corrupted bits would be flipped rather than zeroed out. In this way they add to the variance of the current in the hidden layer neurons. by setting them to zero there is no added variance due to the input noise in the h currents, which artificially aids performance. the larger the hidden layer, the more likely for spurious items to be retrieved, restricting capacity analysis. Was the retrieval for your implementation of the hopfield dynamics done consistently? same for tvt.
- what were the pattern statistics of the stored items?
- it seems that for large stimulus values hopfield degrades the most gracefully among the items compared... why is that?
- what is the loss function being optimized for the hyperparameter learning ?
- 'improves performance for longer sequences' on line 196: by sequence you mean number of stored items? i did not see any reference to temporal dependence in the input-output pairs in 4.2.
- presumably the hidden layer sparsity scaling with number of units is orthogonal to the meta-learning and changes in gating in 4.2
- with the random rewrite version, would you see ever catastrophic forgetting if the number of stored items is larger than N? (in the cycle through neurons version you obviously just overwrite the oldest so forget T-N items)
-missing a bit the biological and/or computational relevance of the metalearning scenario beyond the point that it need not be quite so simple
- biological circuits are almost invariably recurrent with attractor dynamics as a not uncommon dynamical feature; do you see any role for recurrence in this framework (maybe in the context of sequential inputs?)
-for preferrential encoding of salient information not encoding the non-salient information should be the best (because it effectively reduces the number of stored inputs), at least intuitively; not sure how the 'synaptic stability' was implemented exactly, but there are for sure ways to make that work.
- i am somewhat unsure if the comparison to hopfield is really fair in the sense that with key-value one has the extra degree of freedom of the size of the hidden layer. Maybe one way to equate neural resources would be to set N such that the total number of synapses of the two systems is the same (?)
- minor, with the changes in variable naming for hopfield the capacity is o(d) but for KV its o(N) which somehow gets confused in the description.
- since the hopfield net is not designed to do anything other than autoassociative memory its use for some of the more complex tasks seems dubious (and it's unclear how it was set up at all)
- would have liked a more in depth discussion of the relevance of these findings and with biological relevance in mind neural signatures that can differentiate between the different proposals.

Significance: Technically very simple, potentially relevant for interpreting recent data on hippocampal plasticity during behavior.

Originality: Interesting point of intersection between ML and neuroscience.

Clarity: Overall, the text is very clear. In the key learning part the gradual forgetting and link to eq7 needs clarification.


**Time Spent Reviewing:**

6

---

> ### Author Response · Authors · 2021-08-10
> **Addressing comments**
>
> We thank the reviewer for their detailed evaluation of our work. We have addressed each comment below.
>
> - i don't quite understand why the two learning processes need to be different, as long as the softmax is more like a WTA, so the activity in the hidden layer is very sparse h should be able to safely gate
>   - The same synaptic update of eq. 7 could be accomplished without third factors, using a Hebbian rule if the hidden layer activation were one-hot and uncorrelated with the input activity. However, since the hidden layer activity is determined by its weight matrix and the input, there is no simple neuronal mechanism to independently select a hidden neuron for storage.
> However, as suggested by Reviewer dTBY, it may also be possible use \eta^v (t)_ij = q(t) * h'_i(t) instead of  \eta^v (t)_ij = q(t) * \gamma(t) in eq. 9. Correspondingly, this can also be used in eq. 13 as a rapid decay mechanism instead of a passive decay via \lambda_t. The simplest advantage is that local third factors are truly binary, whereas the hidden activity is only approximately one-hot, so the local third factor has a (minor) performance boost. Using the activity h’(t) as a rapid decay mechanism could confer an advantage over the passive decay in eq. 13 while still maintaining biological realism. However, we empirically observe that passive decay (with an appropriately tuned decay factor) operates nearly as well as the upper-bound performance of the local-third-factor-based rapid decay, and is a simpler mechanism.
> - i have some conceptual issues with benchmark 1: in the traditional hopfield net the cue is a valid pattern - in a spin (+/-1) representation the corrupted bits would be flipped rather than zeroed out. In this way they add to the variance of the current in the hidden layer neurons. by setting them to zero there is no added variance due to the input noise in the h currents, which artificially aids performance. the larger the hidden layer, the more likely for spurious items to be retrieved, restricting capacity analysis. Was the retrieval for your implementation of the hopfield dynamics done consistently? same for tvt.
>   - Thank you for pointing out this flaw. We will re-run the analysis with the traditional version of cues. However, in all of our comparisons the inputs were perturbed in the same way, so the networks were all on equal ground.
> - what were the pattern statistics of the stored items?
>   - Binary (+/-1) random (p[+1] = p[-1] = 0.5) and uncorrelated. We will modify the main text to clarify this point, and it is also shown in the Appendix (and the variations for other tasks, e.g. correlated stimuli).
> - it seems that for large stimulus values hopfield degrades the most gracefully among the items compared... why is that?
>   - Note that due to its architecture, the Hopfield network’s input neurons are the same as the output neurons. When it receives a query (i.e. a corrupted input), the output is already initialized with a partial solution (the uncorrupted bits are set to the desired output values). Thus if fraction p of the input bits are corrupted, the chance level is p+(1-p)/2. In contrast, the KEVIN network’s chance level is exactly 50% regardless of the input, since the output units do not have any a priori information about the desired output.
> - what is the loss function being optimized for the hyperparameter learning ?
>   - Thank you for pointing out this missing piece. The loss function is a simple mean-squared error loss. We will add a section to the supplementary material with further details of the model addressing this clarification, as well as others brought up by other reviewers.
> - 'improves performance for longer sequences' on line 196: by sequence you mean number of stored items? i did not see any reference to temporal dependence in the input-output pairs in 4.2.
>   - We will clarify this language. In our case, since items are presented sequentially, the sequence length is equal to the number of stored items.
> - presumably the hidden layer sparsity scaling with number of units is orthogonal to the meta-learning and changes in gating in 4.2
>   - Yes -- ideally there is only one active hidden unit, both in the simplified and meta-learned case.
> with the random rewrite version, would you see ever catastrophic forgetting if the number of stored items is larger than N? (in the cycle through neurons version you obviously just overwrite the oldest so forget T-N items)
>   - No, for the same reason as in the sequential case. The only difference is that some items may be remembered for longer than N by chance, and other items for less than that.
> - missing a bit the biological and/or computational relevance of the metalearning scenario beyond the point that it need not be quite so simple
>   - The main purpose of the meta-learning scenario is to demonstrate that our proposed learning rule can be discovered via optimization. Importantly, it is possible to include additional parameters for more complex learning rules which may further improve the performance of our network.
> - biological circuits are almost invariably recurrent with attractor dynamics as a not uncommon dynamical feature; do you see any role for recurrence in this framework (maybe in the context of sequential inputs?)
>   - There are at least two instances of recurrence that can play a role in our network. First, we suggest that the softmax activation function in the hidden layer can be implemented with recurrent inhibition. By using the softmax, this happens implicitly, but it may be possible to do this with explicit recurrent connectivity among the hidden units. Second, although we have not explored this scenario, it may be possible to run the network recurrently in the autoassociative setting, feeding the output back to the input until it reaches a fixed point (although the existence of such is not guaranteed since the key and value matrices are not symmetric, in contrast to Modern Hopfield Networks).
> for preferrential encoding of salient information not encoding the non-salient information should be the best (because it effectively reduces the number of stored inputs), at least intuitively; not sure how the 'synaptic stability' was implemented exactly, but there are for sure ways to make that work.
>   - The purpose of synaptic stability here is to extend the lifetime of salient information. It is still important to store non-salient information, just not for as long.
> - i am somewhat unsure if the comparison to hopfield is really fair in the sense that with key-value one has the extra degree of freedom of the size of the hidden layer. Maybe one way to equate neural resources would be to set N such that the total number of synapses of the two systems is the same (?)
>   - The decoupling of the input and hidden layer sizes is one of the advantages of our network in comparison with the classic Hopfield model, although for a fair comparison we will re-run our analysis equating the resources in this way. Please note that in our experiments, we use d=N, so our network has exactly twice the number of synapses of the Hopfield net, meaning the scaling is off by at most a factor of 2. This would not change our qualitative results.
> - minor, with the changes in variable naming for hopfield the capacity is o(d) but for KV its o(N) which somehow gets confused in the description.
>   - Thank you for pointing this out, we will fix the notation.
> - since the hopfield net is not designed to do anything other than autoassociative memory its use for some of the more complex tasks seems dubious (and it's unclear how it was set up at all)
>   - We emphasize that while it is possible to use the Hopfield network for heteroassociative tasks by concatenating the key and value, and subsequently perturbing those neurons corresponding to the value, we instead use a network very similar to the Hopfield net, but explicitly designed for heteroassociative memory for this comparison (Bidirectional Associative Memory, Kosko 1988).

---

> > ### Comment · Reviewer_dV7b · 2021-08-11
> > **Post-rebuttal comments.**
> >
> > I thank the authors for the detailed feedback. The reply does not significantly alter my evaluation of the paper, so I will maintain my initial score.

---

### Decision · Program_Chairs · 2021-09-28

**Decision:**

Accept (Poster)

**Comment:**

The significance of the proposal was not satisfactorily established for acceptance. While the committee accepts that the authors' burden is to prove their proposal is an interesting and valid model for biological memory (rather than being competitive with state-of-the-art non-biological memory networks), the experiments were still deemed inadequate in a few different dimensions (see reviewer comments). The metrics for a proposal like this are a bit unclear, but some on the committee would have liked to see scientific merit as evidenced by the proposal being more consistent with actual biological experimental data than past proposals. That was not made clear, and the issue was convoluted by the incomplete analytical comparisons to prior work. For example, it would have helped to place this in context better compared to the sparse distributed memory models of Kanerva et al. from twenty years ago, which seemed very similar, though this proposal uses a more biologically complex soft-max than a step function.  Further, the sequential KEVIN method was not compelling biologically.  Overall, this proposal seemed to fall in the middle of biological plausibility and computationally efficiency, making it not clear how to judge its overall merits and significance, especially with the lack of standard experimental evidence.

**Consistency Experiment:**

NeurIPS has a long history of experimentation. In 2014, NeurIPS ran an experiment in which 10% of submissions were reviewed by two independent committees to quantify the randomness in the review process. This year, we repeated a variant of this experiment to see how the quality of the review process has changed over time.  This paper was part of the experiment and was therefore assigned to two committees (consisting of reviewers, an Area Chair, and a Senior Area Chair) that reached independent decisions.  If both committees made the same recommendation, this recommendation was followed. If a single committee recommended acceptance, the paper was accepted (with the exception of a few cases in which the other committee identified what we considered a fatal flaw, e.g., an error in a key result).

This copy’s committee reached the following decision: **Reject**

The other committee assigned to the paper recommended **Accept (Poster)**.  You can find the other set of reviews, along with any follow up discussion with the authors here:
https://openreview.net/forum?id=6pkC8GUsyDO